# Trade-offs in insect eye nanocoatings: implications for vision, ecology, and climate sensitivity

Mikhail Kryuchkov [1✉], Vladimir Savitsky [2], Marc Jobin[3], Stanislav Smirnov[4,5], Mirza Karamehmedović [6], Jana Valnohova[1] & Vladimir L Katanaev [1✉]

## Abstract

**Functional traits shape ecological niches, yet the interplay between nanoscale structural modifications, sexual dimorphism, and habitat range remains poorly understood. In fireflies, cuticular nanostructures that enhance bioluminescent signaling efficiency also impose ecological constraints. Anti-reflective nanocoatings improve cuticle transparency and optical performance but typically increase surface adhesion, reducing fitness. In *Luciola lusitanica*, this trade-off is mitigated by temperature-sensitive nanocoatings that form only within a narrow thermal range, limiting habitat expansion. This study presents the first thermodynamic analysis of environmentally constrained nanocoating formation, demonstrating how small temperature fluctuations can destabilize protein-lipid self-assembly. These findings link nanoscale biophysics to ecological resilience, providing a framework to understand how the environmental sensitivity of structural self-organization shapes adaptation, species distribution, and evolutionary potential.**

**Keywords** Reaction-Diffusion; Nanostructures; Sexual Dimorphism; Ecological Resilience; Thermodynamic Analysis
**Subject Category** Evolution & Ecology

## Introduction

Animal communication is shaped by a dynamic interplay of selective pressures, including both natural and sexual selection. These forces have driven the evolution of diverse signaling modalities (visual, olfactory, auditory, and tactile) that mediate courtship, territorial defense, group cohesion, and predator avoidance (Hauser and Konishi, 1999). Among insects, fireflies have evolved one of the most striking visual communication systems: the use of bioluminescent flashes for mate recognition and attraction (Martin et al, 2019). The effectiveness of such light-based communication depends not only on the production and reception of species-specific signals but also on the optical properties of the body structures through which these signals pass. Before reaching the retina, light emitted from the photocytes of the firefly lantern must traverse two cuticle-air interfaces, where substantial reflection can occur. To minimize light losses, insects have evolved nanostructured cuticular coatings with features smaller than the wavelength of emitted light. These subwavelength nanocoatings create a gradual refractive index transition between the cuticle and air, reducing Fresnel reflections and enhancing light transmission efficiency (Kryuchkov et al, 2017a; Raut et al, 2011; Stavenga et al, 2005).

Anti-reflective nanocoatings occur naturally across a range of organs (Deinega et al, 2011; Stavenga et al, 2005; Wan and Gorb, 2023), including wings (Ivanova et al, 2017; Wilts et al, 2017), eyes (Gemne, 1971; Meyer-Rochow and Stringer, 1993), and firefly lanterns (Kryuchkov et al, 2021; Qarony et al, 2018; Siddique et al, 2015). Specialized corneal nanocoatings have been described in beetles (Kryuchkov et al, 2025), moths (Dewan et al, 2012; Kryuchkov et al, 2017a), mosquitos (Gao et al, 2007) and fruit flies (Kryuchkov et al, 2020) and can vary even between regions of a single compound eye (Blagodatski et al, 2014; Kryuchkov et al, 2017c). Corneal nanostructures are precisely adapted to optical function and habitat, with altered geometry reducing visual efficiency and fitness. Across Lepidoptera and Diptera, nipple array dimensions strongly affect reflectance and transmittance (Stavenga et al, 2005). In *Drosophila melanogaster*, Retinin downregulation shortens corneal protrusions two- to threefold (Kryuchkov et al, 2020), impairing phototaxis and shortening lifespan (Hodge et al, 2022). Domesticated *Bombyx mori* lacks such nanostructures compared to wild-type *Bombyx mandarina*, reflecting relaxed selection (Kryuchkov et al, 2017b). Ecological demands also drive eye-region specialization, as seen in Gyrinidae and Ascalaphidae, where dorsal and ventral surfaces differ for aerial versus aquatic or visible spectra versus UV-based vision (Blagodatski et al, 2014; Kryuchkov et al, 2017c).

A key factor constraining nanocoating evolution is the trade-off between optical and mechanical surface properties. Cuticular nanostructures are typically optimized for either anti-reflective or anti-adhesive functions, but rarely both (Kryuchkov et al, 2020). Prominent nanocoatings enhance optical transmission but increase

[1]Department of Cell Physiology and Metabolism, Faculty of Medicine, University of Geneva, Rue Michel Servet 1, CH-1211 Geneva, Switzerland. [2]Zoological Museum of the Lomonosov Moscow State University, Bol'shaya Nikitskaya Street 2, 125009 Moscow, Russian Federation. [3]HEPIA, University of Applied Sciences of Western Switzerland (HES-SO), 4 Rue de la Prairie, CH-1202 Geneva, Switzerland. [4]Section of Mathematics, University of Geneva, Rue du Conseil-Général 7-9, CH-1205 Geneva, Switzerland. [5]Department of Mathematics and Computer Science, St. Petersburg University, 29 Line 14th, Vasilyevsky Island, 199178 St. Petersburg, Russian Federation. [6]Department of Applied Mathematics and Computer Science, Technical University of Denmark, DK-2800 Kgs Lyngby, Denmark. ✉E-mail: Mikhail.Kryuchkov@unige.ch; Vladimir.Katanaev@unige.ch

surface adhesion, reducing the insects ability to self-clean through grooming and thereby elevating the risk of contamination or infection (Zhukovskaya et al, 2013). This trade-off defines a narrow adaptive window for insects relying simultaneously on efficient visual signaling and surface cleanliness.

These opposing pressures are particularly evident in fireflies of the genus *Lampyris*, where flying males locate sedentary, glowing females (De Cock, 2009). In this mating system, anti-reflective nanocoatings are expected on female lanterns and male eyes, whereas the rest of the cuticle remains anti-adhesive. In contrast, *Luciola* fireflies, which employ bidirectional signaling between males and females (Papi, 1969), face a different challenge: maintaining optical efficiency while preventing fouling of illuminated surfaces. This behavioral shift may have driven the evolution of hybrid nanocoatings with both anti-reflective and self-cleaning properties, a rare solution balancing the conflicting demands of communication and surface maintenance.

## Results and discussion

### Paving-stones-like nanocoatings combine anti-reflectivity with low wettability without compromising mechanical resilience

Comparative analysis of beetle corneal nanocoatings (Appendix Table S1) (Blagodatski et al, 2014; Blagodatski et al, 2015; Kryuchkov et al, 2025; Mishra and Meyer-Rochow, 2006), revealed pronounced species-specific differences and in some cases sexual dimorphism (Fig. 1A,B). In *Lampyris noctiluca*, males possess prominent nipple-like nanostructures on their eyes, reflecting their reliance on vision during mate searching, whereas females have anti-reflective coatings localized to their lanterns. By contrast, both sexes of *Luciola lusitanica* exhibit pronounced nanocoatings across all surfaces, with a distinctive paving-stone-like morphology on the cornea, suggestive of a specialized adaptation for bioluminescent communication (Fig. 1C,D).

To expand the comparative framework, we included three additional species. *Lampyris orientalis* displays sexual dimorphism closely resembling *L. noctiluca* (Fig. EV1A). *Aquatica lateralis*, behaviorally similar to *L. lusitanica* (Ohba, 2004), possesses pronounced nanocoatings on both eyes (Fig. EV1B) and lanterns (Kim et al, 2012; Kryuchkov et al, 2021) but no novel nanostructure types. The non-luminescent *Cantharis rustica* shows smooth, dimpled corneal surfaces (Fig. EV1C).

Spectroscopic analysis confirmed that anti-reflective efficiency correlates with nanostructure height (Kryuchkov et al, 2017c). Reflectance was similarly low across the visual spectrum in *L. lusitanica* males and females, and in *L. noctiluca* males, whereas *L. noctiluca* females, with less developed nanocoatings, reflected two- to threefold more light (Fig. 2A). This indicates a nanoscale adaptation aligned with species-specific mating strategies.

We next evaluated corneal wettability. Two-dimensional adhesion force mapping (Devi et al, 2016) revealed that smaller, maze-like nanostructures exhibit lower average adhesion than larger morphologies (Figs. 2B and EV2B). In *L. noctiluca*, females prioritized hydrophobicity (contact angle ≈ 80°) over optical sensitivity, whereas males nanostructures were more hydrophilic (≈60°). The other examined species (*L. orientalis*, *A. lateralis*, *C. rustica*) followed a near-linear relationship on a simulated transmittance/contact angle plot, consistent with the expected trade-off between anti-reflective and self-cleaning properties (Fig. EV1D). While optical simulations reproduced the general trend, the absolute values were considerably lower, compared to direct measurements. This discrepancy may arise from the complex curvature of natural corneal surfaces. Alternatively, it may indicate that optical effects in insect cuticles stem not only from surface corrugation but also from the nanoscale organization of molecular constituents (Lukosz, 1979; Ràfols-Ribé et al, 2023; Tewarson, 1967; Wasey et al, 2000).

Remarkably, *L. lusitanica* deviated from this trend: despite pronounced nanostructures and relatively high adhesion, its cornea maintained high hydrophobicity (Figs. 2C and EV1D). Analysis suggests that air entrapment within narrow gaps of the paving-stone-like pattern (~10% surface coverage) enhances water repellency, similar to the mechanism observed in mosquitoes (Gao et al, 2007), thereby enabling simultaneous anti-reflectivity and self-cleaning (Fig. EV2).

Such cases of dual-functionality remain rare, but some examples exist. *Papilio* corneal nanostructures are chemically hydrophobic, providing both water repellence and optical transparency (Wang et al, 2025), and mosquito corneal protrusions achieve a similar dual efficiency (Gao et al, 2007). The formation of such multifunctional architectures appears constrained by oscillatory assembly dynamics, which restrict the parameter space for stable pattern formation (Kryuchkov et al, 2022; Wang et al, 2025).

The case of paving-stone-like nanocoatings raises the question of structural limitations. One hypothesis is that mechanical softness might restrict the evolution of such patterns. However, nanoindentation measurements of Young's modulus demonstrate that *L. lusitanica* nanocoatings are at least as mechanically robust as the nipple-like structures of *L. noctiluca* males and exceed the stiffness of female maze-like patterns (Figs. 2D and EV2C). These observations challenge the presumed trade-off between anti-reflectivity and hydrophobicity, revealing a structurally resilient nanocoating capable of optimizing both functions.

### Mathematical modeling reveals fragility of paving-stones-like nanocoating formation

To understand why *Luciola lusitanica* evolved a unique bifunctional nanocoating, we analyzed the self-assembly mechanism underlying corneal nanostructure formation using a reaction-diffusion framework. In this model, nanostructures arise from the interplay between a slowly diffusing protein activator and a rapidly diffusing lipid- or wax-like inhibitor (Kryuchkov et al, 2020; Turing, 1952). During cuticle formation, these molecular interactions generate a periodic pattern within the first thin layer of the corneal cuticle, upon which subsequent layers are deposited (Gemne, 1971). The emergence of these spatial gradients in protein and lipid concentration is critical for pattern initiation (Kryuchkov et al, 2020). Our mathematical model reproduces the essential dynamics of this process while simplifying the biochemical complexity (Fig. 3A,B).

Previous theoretical studies have successfully modeled the formation of nipple- and maze-type corneal nanostructures (Blagodatski et al, 2015; Kondo and Miura, 2010; Kryuchkov et al, 2020; Kryuchkov et al, 2024), but the mechanism generating

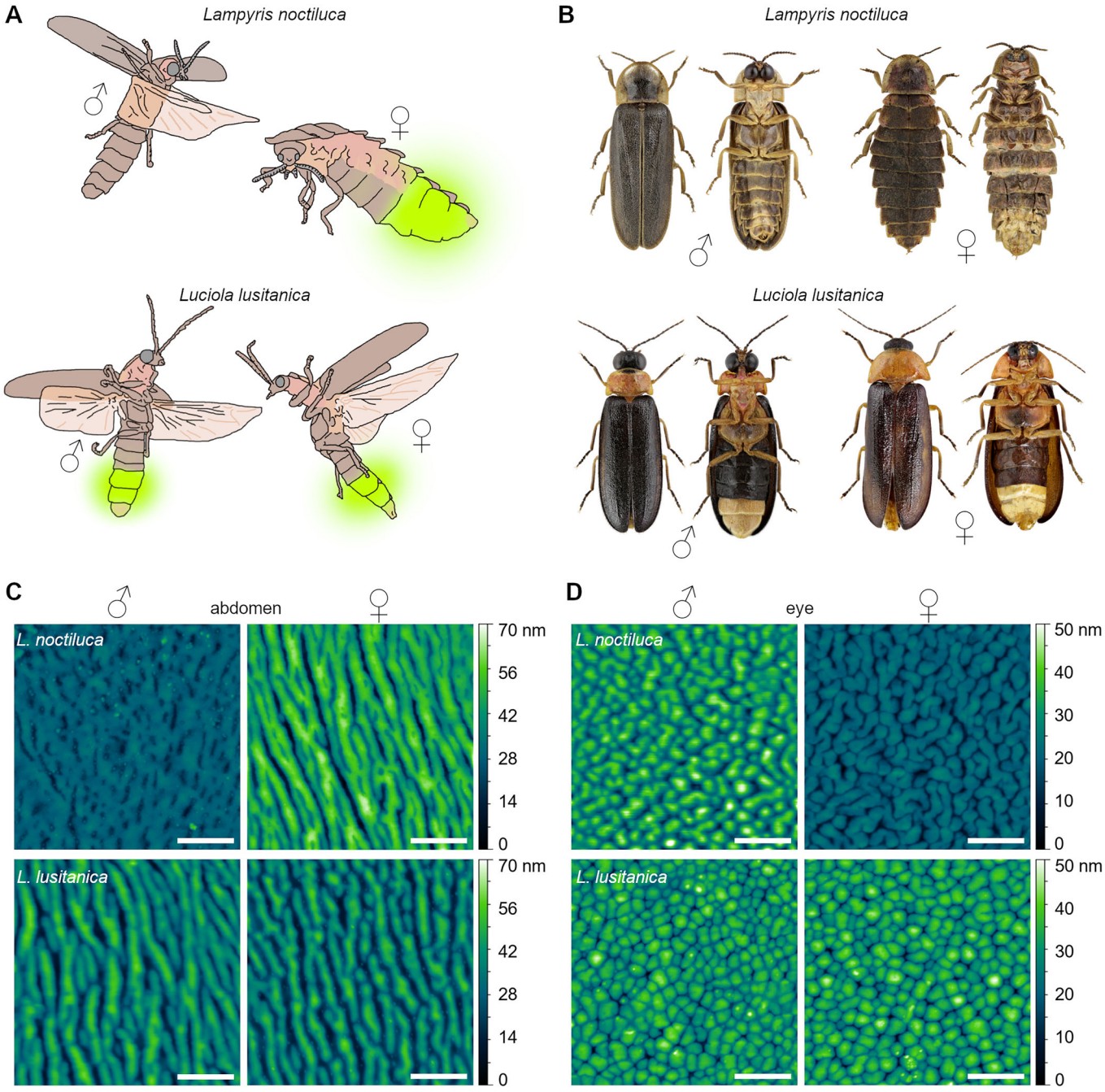

**Figure 1. The sexual behavior of fireflies affects their structures on the macro- and nano-levels.**

(A) Sexual behavior of different fireflies. Non-flying females of *L. noctiluca* attracting flying non-emitting light males, whereas males and females of *L. lusitanica* both emit light and search for the signals. (B) Light microscopy images showing the dorsal and ventral views of fireflies. A clear distinction between *L. noctiluca* male and female could be observed, unlike for *L. lusitanica*. (C, D) Representative AFM (Atomic Force Microscopy) scans of the fireflies' lanternal abdomen surfaces and corneal eye surfaces. Differences between female and male nanostructural coatings can be observed for *L. noctiluca* (upper row) but not for *L. lusitanica* (bottom row). Surface height is indicated by the color scale shown next to the images. Scale bars: 1 μm. Source data are available online for this figure.

the characteristic paving-stones-like morphology of *L. lusitanica* remained unresolved. Analytical treatment shows that the nanostructure breadth ($\omega$) depends on the diffusion coefficient ($D$) and the reaction rate ($f$) of the morphogens, following the relationship: $D \propto f\omega^2/4\pi^2$ (Kondo and Miura, 2010). Because *L. lusitanica* nanostructures are markedly narrower than those of *L.*

*noctiluca*, the underlying morphogens must either diffuse more slowly or react faster (Fig. EV3A,B). Numerical simulations revealed that only a reduction in activator diffusion reproduces the observed paving-stones-like geometry, as decreased mobility prevents neighboring units from merging (Blagodatski et al, 2015; Buscher et al, 2018). Remarkably, a two-fold reduction in the

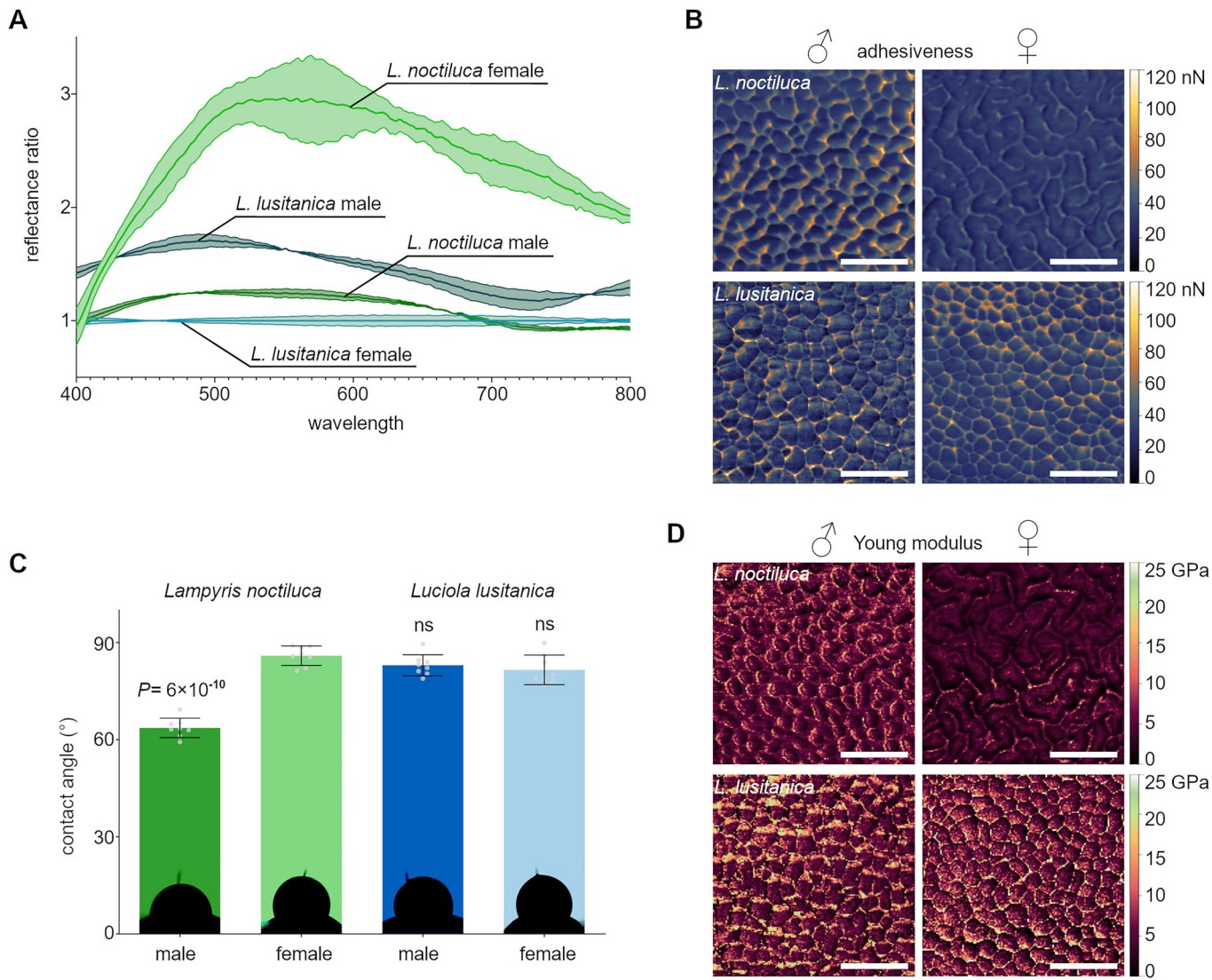

**Figure 2. Physical properties of investigated corneal nanocoatings.**

(A) Ratio of reflection spectra measured for the four corneal nanocoatings, normalized to the average of *L. lusitanica* females; the resulting spectra are presented as a reflectance ratio; mean ± SEM; $n = 5$ (individual eyelets), $N = 2$ (biologically independent animals). (B) Representative adhesion force scans of investigated corneal nanocoatings. Only the maze-like nanostructures (upper right panel) exhibit reduced average adhesion. (C) Contact angle of water droplets on the corneal cuticles of the fireflies. Actual representative images of the droplets are shown as bar inserts (in black). Data represent mean ± SD, $n = 8$, $N = 2$ (biologically independent animals). Statistical significance (t-test) is provided in comparison to the contact angles of *L. noctiluca* female corneae; "ns" indicates non-significant ($P > 0.05$). (D) Representative Young modulus scans of investigated corneal nanocoatings. Scale bars: 1 μm. Source data are available online for this figure.

activator diffusion coefficient was sufficient to transform nipple-like patterns into paving-stones-like arrays (Fig. EV3C,D; Appendix Table S2).

Further simulations delineated the parameter space within which each pattern type can form. Nipple- and maze-like nanocoatings emerged robustly across a broad range of reaction-diffusion ratios, whereas the paving-stones-like configuration appeared only within a narrowly defined window (Fig. 3C; Appendix Fig. S1). This suggests that although *L. lusitanica* nanocoatings confer superior optical and self-cleaning properties, their assembly is highly sensitive to environmental perturbations, potentially explaining their restricted phylogenetic and geographic distribution.

Because reaction-diffusion systems are strongly temperature-dependent (Carballido-Landeira et al, 2010; McIlwaine et al, 2009), we next tested whether thermal fluctuations could disrupt nanostructure formation. The self-assembly of corneal nanocoatings depends on the delicate balance between reaction kinetics and molecular diffusion (Fuseya et al, 2021; Kondo and Miura, 2010; Kryuchkov et al, 2020; Turing, 1952). Temperature affects both processes, typically accelerating chemical reactions while increasing molecular mobility. Even small variations can shift this balance, leading to altered pattern periodicity or transitions between morphologies (Carballido-Landeira et al, 2010; Ritchie, 2018).

We modeled temperature dependence using the Arrhenius equation, $k = Ae^{\frac{-E_a}{RT}}$, incorporating activation energies ($E_a$) for

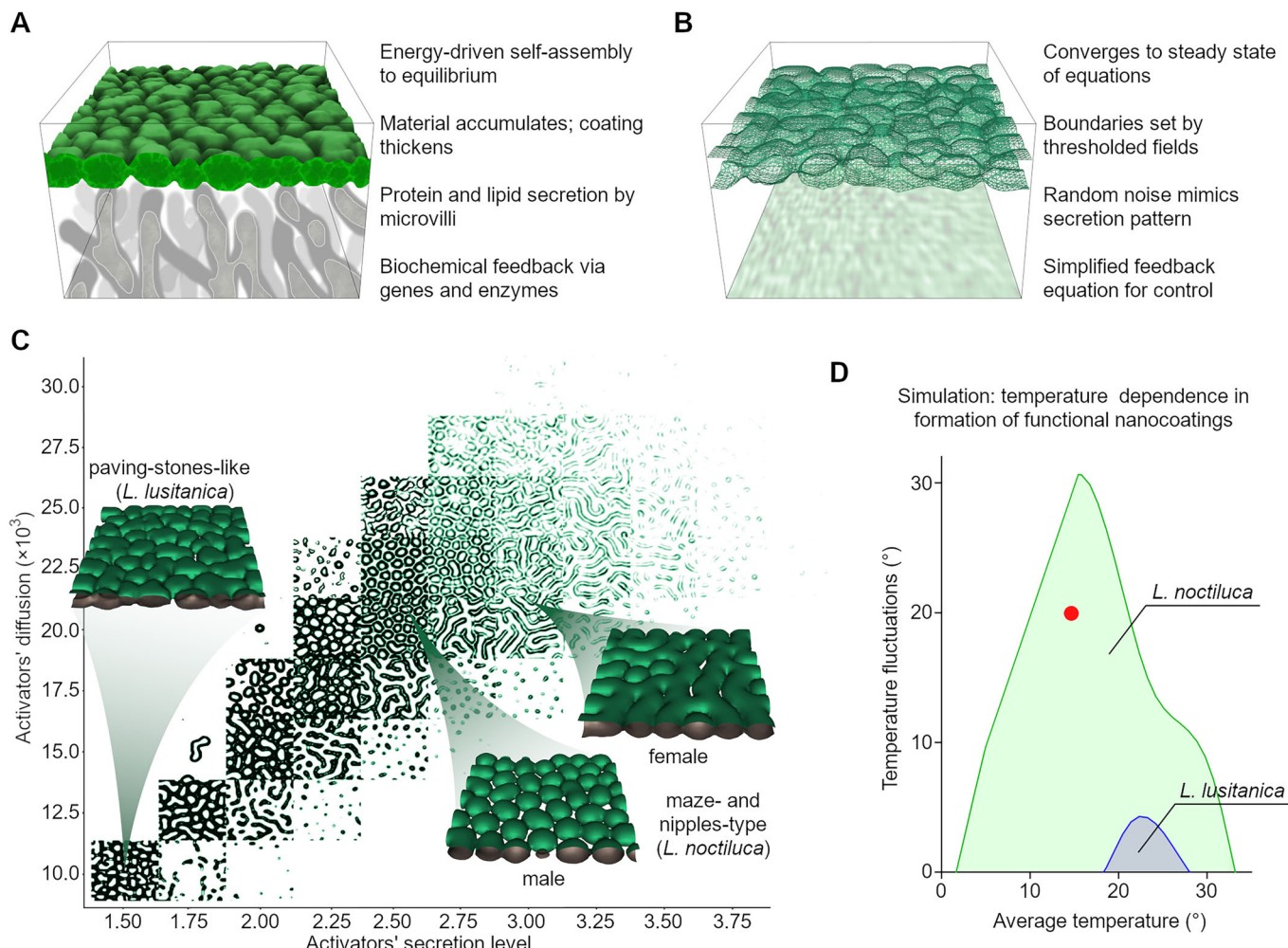

**Figure 3. Thermal sensitivity of paving-stones-like nanostructures.**

(A, B) Assembly process of the corneal nanostructures, observed in vivo (A) and simulated in silico (B). In (A), a real image of *L. lusitanica* corneal nanocoatings (green) is superimposed on the artistically visualized cone cells' microvilli (gray, inspired by (Bernhard and Miller, 1962)). Parameters used for the simulation in (B) are provided in Appendix Table S3. (C) Parameter space defined by the Turing activator secretion and diffusion values illustrating the region of Turing instability. Whereas nipple- and maze-like structures dwell well inside the instability region, the paving-stones-like structures can be formed only within a strictly insular area. Parameters directly neighboring this narrow area lead to loss of instability. (D) Simulations show that *L. noctiluca*-like nanostructures can be formed in a broad range of temperatures (green), but the conditions for paving-stones-like (*L. lusitanica*) structures are much narrower (blue). Source data are available online for this figure.

diffusion, reaction, and secretion processes (Carballido-Landeira et al, 2010; Serna et al, 2017). Meta-analyses indicate that diffusion has typically a mean activation energy of 30 MJ/mol, reactions require 65 MJ/mol, and secretion processes vary based on the molecule involved (proteins: 90 MJ/mol; wax: 115 MJ/mol) (Craig, 1979; Lee and Lentz, 1998; Ritchie, 2018; Small, 2003). Due to the biochemical synthesis, fast temperature shifts have delayed effects on a secretion, postponing consequences by ~2 h for proteins and ~2.5 h for waxes (Bonnemaison, 2014).

Model integration revealed that *L. lusitanica* nanocoating formation is markedly more temperature-sensitive than that of *L. noctiluca* (Fig. 3D; Appendix Table S3). Simulations mimicking springtime temperature cycles identified periods when *L. lusitanica* structures fail to assemble, while *L. noctiluca* coatings remain stable (Fig. EV4A,B; Appendix Fig. S2). This provides a mechanistic basis for *L. lusitanica*'s restricted habitat, showing that its bi-functional

nanocoating is environmentally fragile. The results suggest that thermal sensitivity constrains the evolution of such nanostructures, a hypothesis that may extend to other arthropods. Direct empirical evidence linking temperature fluctuations to changes in arthropod nanostructures remains scarce. One of the few examples comes from the collembolan *Cryptopygus clavatus* whose cuticular nanostructures shift from convex in winter-acclimated individuals to concave in summer-acclimated ones, reducing surface hydrophobicity from a contact angle of 166° to 140° (Gundersen et al, 2015). In *Polyommatus icarus* pupae, prolonged cold stress alters wing coloration, likely through temperature-dependent changes in the size of photonic nanoholes (Kertész et al, 2017; Piszter et al, 2019). Similarly, rearing temperature in the guava fruit fly *Bactrocera correcta* affects the number of nanopores in olfactory sensilla, correlating with reduced odor sensitivity (Guo et al, 2022).

## In vitro recapitulation of temperature-sensitive paving-stones-like nanocoatings

To experimentally validate the temperature-dependent formation of paving-stones-like nanostructures, we adapted an in vitro system that mimics insect corneal nanocoatings. Previous studies have shown that co-assembling wax-like inhibitors with Retinin, a Turing activator from Drosophila, produces a range of bioinspired nanostructures, including nipple-, maze-, and dimple-like patterns, each with distinct optical and hydrophobic properties (Kryuchkov et al, 2020). Mathematical modeling suggested that reducing diffusion coefficient of Retinin could induce paving-stones-like nanocoatings formation.

To test whether reducing the diffusion coefficient of the Turing activator could induce the formation of paving-stone-like nanostructures, we engineered a Retinin and NanoLuciferase (England et al, 2016) fusion protein (RetLuc; 38 kDa, 357 amino acids) and compared it with the native Retinin (19 kDa, 179 amino acids) (Appendix Fig. S3). Single-particle tracking by TIRF microscopy confirmed that RetLuc diffused significantly more slowly than Retinin ($1.34 \pm 0.20 \times 10^{-8}$ vs. $2.34 \pm 0.34 \times 10^{-8}$ cm²/s), achieving the predicted two-fold reduction in diffusion coefficient. As expected, the Turing inhibitor displayed a diffusion coefficient of $7.29 \pm 0.32 \times 10^{-8}$ cm²/s, approximately 3–5 times higher than that of the Turing activators (Fig. EV5A), consistent with reaction-diffusion model requirements (Pearson and Horsthemke, 1989). Despite this change in diffusivity, RetLuc retained Retinin's binding kinetics toward wax-like molecules, displaying comparable lipid-binding and folding behavior, as confirmed by wave-guide interferometry (Fig. EV5B). Both proteins self-assembled into nanocoatings on glass surfaces; however, RetLuc produced narrower, more sharply defined nanostructures, whereas Retinin formed broader, less compact patterns (Figs. 4A,B and EV5C). Knowing the precise diffusion coefficients allowed us to plot the relationship between $D$ and $f$ parameters, indicating that the reaction rates of both proteins remain within the same range of a reaction rate (Fig. EV5D), consistent with previous theoretical predictions (Kryuchkov et al, 2020; Kryuchkov et al, 2024).

Additionally, the RetLuc-based nanocoatings exhibited increased hydrophobicity (Fig. 4C,D), closely resembling *L. lusitanica* natural corneal nanocoatings, while Retinin-based coatings mirrored those of *L. noctiluca*.

To further investigate their biophysical properties, we experimentally assessed the temperature sensitivity of Retinin- vs. RetLuc-based nanocoatings. Unlike insect corneal structures, formation of the artificial nanocoating depends on morphogen concentration and surface adsorption rather than secretion. Using the Arrhenius equation, we modeled protein and wax adsorption kinetics, assigning activation energies of 30 kJ/mol for wax adsorption and 55 kJ/mol for protein adsorption (Fritz et al, 2021), successfully replicating the experimentally observed nanocoatings (Fig. 4E,F).

Both simulations and experiments revealed a temperature-dependent constraint for RetLuc: stable nanocoatings formed only within a narrow temperature range (25–30 °C). At lower temperatures (<25 °C), no RetLuc-based nanostructures assembled, while at higher temperatures (>30 °C), uncontrolled nanostructure growth

in solution led to adhesion and aggregation (Fig. 4E (brown structures), Fig. 4F). In contrast, Retinin maintained robust nanocoating formation across the entire tested range (15–35 °C, Appendix Fig. S4).

The luciferase activity of RetLuc enabled real-time assessment of nanocoating integrity. Even after washout and drying, RetLuc retained enzymatic activity, with peak luminescence at 25 °C, corresponding to the highest level of immobilized protein (Fig. 4G,H). These findings establish RetLuc as a bioinspired model for *L. lusitanica* nanocoatings and highlight the fragility of paving-stones-like nanostructures under temperature fluctuations, compared to the more resilient nipple-like structures.

## Habitat limitation and consequences of temperature sensitivity

Our analyses indicate that the formation of *L. lusitanica* corneal nanostructures is restricted to mild climatic conditions with minimal temperature fluctuations. Examination of historical collections (Kazantsev, 2010) (1886–2022, Appendix Table S4) revealed that *L. lusitanica* occupies mainly temperate Mediterranean zones and avoids elevations above 2000 m, whereas *L. noctiluca* ranges broadly across Europe, including high altitudes (Fig. 5A; Appendix Table S4) (De Cock, 2009; Gardiner and Didham, 2020; Kazantsev, 2010; Meyer-Rochow, 2009; Novák and De Cock, 2017; Papi, 1969).

These observations suggest that *L. lusitanica* populations may exhibit structural plasticity in atypical environments, such as high elevations. Among the available material, we identified only two *L. lusitanica* males originating from elevations between 1900 and 1950 m. Wettability measurements in one (* in Appendix Table S4) of these specimens revealed a bimodal distribution of contact angles (≈85° and ≈60°), indicating a mosaic eye surface (Fig. 5B, purple bar). AFM analysis confirmed the coexistence of the typical paving-stones-like nanostructures with nipple-like elements (Fig. 5C), including intermediate transition zones (Appendix Fig. S5). These regions exhibited higher adhesion (Fig. 5D), likely accounting for the reduced contact angles. Together, these data reveal temperature-dependent variation in nanostructure formation.

While the observed changes of nanostructures are modest, our reaction-diffusion simulations predict that a further temperature change could destabilize nanocoating assembly entirely. As nanocoatings form before other cuticular layers (Ando et al, 2019; Gemne, 1971; Kryuchkov et al, 2017a), such disruption may impair cuticularization, potentially resulting in non-viable phenotypes (Ghosh and Treisman, 2024). Thus, nanoscale adaptations that enhance optical or self-cleaning performance can also impose tight environmental constraints on development (Pérez-Ramos et al, 2019; Tang et al, 2024; Wan and Gorb, 2023).

Simulations show that even a 0.2 °C temperature rise could push the system beyond its stability threshold. Similar threshold effects occur in other biological contexts, where small temperature increases trigger major phenological or developmental shifts (Beil et al, 2021; Chamberlain et al, 2019; Meier et al, 2018; Vitasse et al, 2025; Zohner et al, 2020). Such sensitivity underscores how minor environmental changes can surpass local kinetic or physiological limits, leading to failure in systems adapted to narrow conditions.

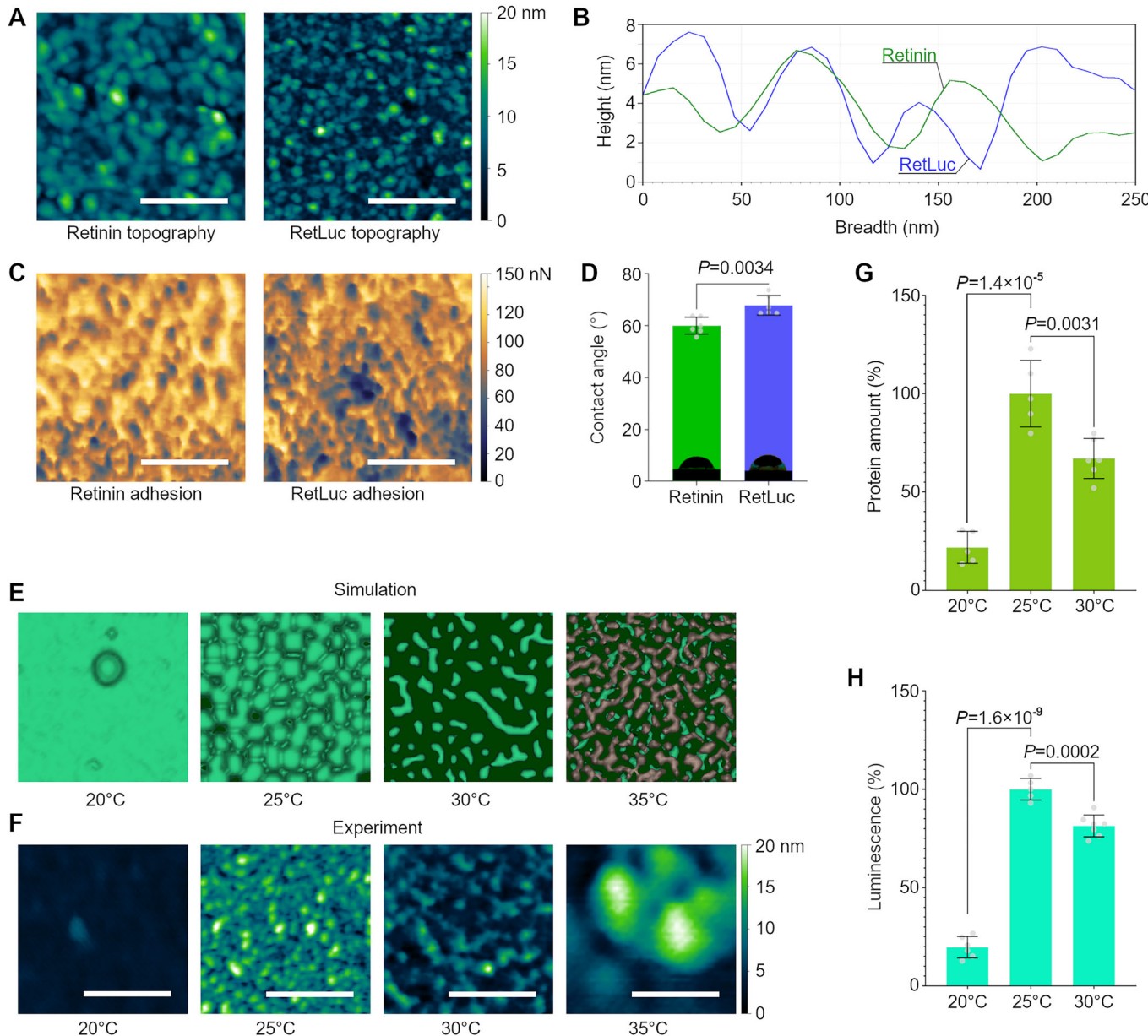

**Figure 4.  Artificial nanocoatings mimic naturally occurring nanocoatings.**

(A) Representative AFM scans of the artificial nanocoatings showing their topography. (B) Representative cross-sections of artificial nanocoatings show that the RetLuc-based nanostructures (blue line) are narrower and have tighter spaces between them than the Retinin-based ones (green line). (C) Adhesion force microscopy of Retinin- and RetLuc-based nanocoatings reveals a different ratio of highly to weakly adhesive regions, resulting from narrower contact areas and a denser packing of protrusions. (D) The contact angle of water droplets on the obtained nanocoatings. Actual representative images of droplets are shown as inserts in bars. Data are shown as mean ± SD. *P*-value (D) determined by two-tailed nonparametric t-test. (E, F) RetLuc-based nanocoatings simulated (E) and experimentally observed (F) at temperatures varying from 20 to 35 °C reveal drastic changes in the topography of the resulting artificial nanocoatings, from smooth surfaces to micrometer-sized aggregates. (G) For the RetLuc-based nanocoatings formed at different temperatures, the total amount of protein remaining after wash-out correlates with the luminescence data in Fig. 4H. The protein levels are quantified by SDS-PAGE and normalized to the levels measured at 25 °C. The data are presented as mean ± SD, $n = 5$ (20 and 25 °C), and 6 (30 °C), $N = 2$ (independent experiments), statistical analysis with a two-tailed t-test. (H) RetLuc-based nanocoatings maintain the enzymatic (luciferase) activity. Data are normalized to the levels at 25 °C. The data are presented as mean ± SD, $n = 6$ (20 °C), 5 (25 °C), and 7 (30 °C), $N = 2$ (independent experiments), statistical analysis with a two-tailed t-test. Scale bars: 0.5 μm. Source data are available online for this figure.

With global warming likely underestimated (Supran et al, 2023) and mitigation limited (Armstrong McKay et al, 2022), these thresholds may soon be crossed in nature. The paving-stones-like nanocoatings, though optically and hydrophobically efficient, exemplify how evolutionary success depends more on robustness than maximal performance, a pattern broadly observed across taxa (Cope, 1904; Goessling et al, 2025; Goulson et al, 2015; Spalding and Brown, 2015).

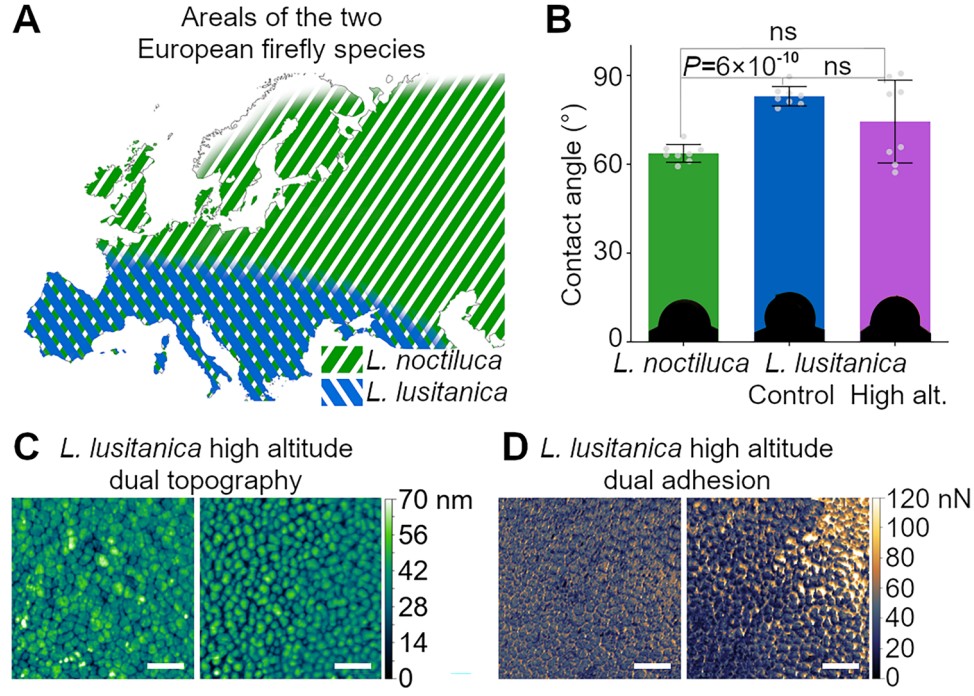

**Figure 5. *L. lusitanica*, from sub-optimal conditions, forms two types of corneal nanostructures.**

(A) Areals of the two firefly species hint that *L. lusitanica* is more sensitive to environmental variability than *L. noctiluca*. (B) The contact angles of water droplets on the corneal cuticle of *L. lusitanica* specimen collected at high altitude indicate the simultaneous presence of hydrophobic and hydrophilic surface properties. The inserts in the bars display actual representative images of the droplets (in black). The data are presented as mean ± SD, with a sample size of $n = 8$. The notation "ns" indicates the absence of significant differences. (C, D) Representative AFM scans of the corneal eye surfaces of *L. lusitanica* from high altitude, demonstrating their topography (C) and adhesion (D). High altitude nanocoatings defects become more evident when examining the adhesion of these structures (D). All data related to L. lusitanica from the high-altitude site were obtained from a unique specimen ($N = 1$), in other cases $N = 2$. Scale bars: 1 μm. Source data are available online for this figure.

# Methods

### Reagents and tools table

| Reagent/Resource | Reference or Source | Identifier or Catalog Number |
|---|---|---|
| **Experimental models** | | |
| *Lampyris noctiluca* (11 ♂♂, 11 ♀♀) | Priosko-Terrasny Nature Reserve, Moscow Region. June 2016; June 2018; and June 2019. | |
| *Lampyris orientalis* (3 ♂♂, 2 ♀♀) | Abkhazia, Bzyb Range near Sukhum, 520–2100 m. July 2014–2017 | |
| *Luciola lusitanica* (6 ♂♂, 2 ♀♀) | Western Caucasus, Tuapse District, Sochi area, Adygea, Abkhazia, 0–1950 m. June and July 1964–2014 | – |
| *Aquatica lateralis* (1 ♂) | Kunashir Island, Sakhalin Region. August 1988 | |
| *Cantharis rustica* (3 specimens) | Priosko-Terrasny Reserve. June 2017 | |
| BL21(DE3)pLysS *E. coli* strain | Merck | 69451-3 |
| **Recombinant DNA** | | |
| Go1-CASE | Katanaev laboratory | |
| pET23Retinin | Katanaev laboratory | |

| Reagent/Resource | Reference or Source | Identifier or Catalog Number |
|---|---|---|
| **Oligonucleotides and other sequence-based reagents** | | |
| PCR primers | Sigma | |
| **Chemicals, Enzymes and other reagents** | | |
| Restrictases | NEB | |
| NHS | Xantec | |
| EDC | Xantec | |
| Ethanolamine | Merck | 398136 |
| Decanoic acid | Merck | W236403-K |
| Carnauba wax #1 | Aldrich Chemistry | 243213 |
| Sodium borate | Merck | 31457 |
| NaCl | Merck | S9888 |
| SDS | Merck | L3771 |
| Glass coverslip | epredia | 22×22-1-002G |
| 96-well black microplate | Greiner bio-one | 655076 |
| Furimazine | Chem Shuttle | 185252 |
| Tris-HCl | Merck | 10812846001 |
| Glycerol | Merck | G5516 |

| Reagent/Resource | Reference or Source | Identifier or Catalog Number |
|---|---|---|
| β-mercaptoethanol | Merck | M3148 |
| Bromophenol blue | Merck | 114391 |
| ATTO565 NHS-ester | ATTO-TEC GmbH | |
| BODIPY™ 500/510 C$_1$, C$_{12}$ | ThermoFisher Scientific | D3823 |
| IPTG | Merck | I6758 |
| **Software** | | |
| Gwyddion 2.55 | https://gwyddion.net/ | |
| Ready 0.8 | https://github.com/GollyGang/ready | |
| Mathematica 13.3 | Wolfram Research | |
| ImageJ | https://imagej.net/ij/ | |
| GraphPad Prism 10.0.3 | https://www.graphpad.com/features | |
| **Other** | | |
| XE-100 | Park Systems | |
| NX7 | Park Systems | |
| NX10 | Park Systems | |
| PPP-NCHR cantilevers | Nanosensors | |
| Tap300AI-G cantilevers | Budget Sensors | |
| QE Pro spectrometer | Ocean Insight | |
| Zeiss AxioScope A1 | Zeiss | |
| Zeiss Epiplan-Apochromat 50x objective | Zeiss | |
| Measurement fiber 230 μm core | Ocean Insight | |
| Micromanipulator MN-4, MMO-203 | Narishige | |
| Axiovert 40 C | Zeiss | |
| Cell press | Constant Systems | |
| HisPur Ni-NTA resin | ThermoFisher Scientific | 88221 |
| Superdex™ 200 Increase 10/300GL FPLC column | Cytiva | GE28-9909-44 |
| Creoptix WAVE delta | Creoptix AG | |
| 4PCH chips | Creoptix AG | |
| Bandelin sonorex | Bandelin electronic | |
| M Plex | Tecan | |
| Nikon Eclipse Ti microscope | Hamamatsu | |

## Coleoptera sample preparation

The cuticles were gently cleaned with milli-Q water, and the resultant samples were attached to a coverslip by a double-sided tape.

## Atomic force microscopy (AFM)

Topography data were collected in tapping mode using an XE-100 microscope and PPP-NCHR cantilevers (radius of curvature < 25 nm), at the Micronanotechnology facility (HES-SO, Geneva, Switzerland). Young's modulus and Adhesion force measurements were performed using NX7 and NX10 microscope and Tap300AI-G cantilevers, with the nominal spring constant k = 40 N/m and curvature radius 10 nm, at Adolphe Merkle Institute (Fribourg, Switzerland). Measurements were done in the PinPoint™ mode, with the following parameters: speed 30 μm/s and set point 290 nN. Quantification of the elastic modulus in this mode was based on the Hertzian model. All scans were initially measured with a resolution of 256 × 256 pixels. Gwyddion software was used for the visualization and quantitative analysis (mean adhesion force, Young modulus, roughness ratio).

## Reflectance measurements

Measurements were performed with QE Pro spectrometer, connected to a custom-adapted Zeiss AxioScope A1, at Adolphe Merkle Institute (Fribourg, Switzerland). The measurement fiber was placed confocal to the image plane, resulting in an effective measurement area of 8 μm.

## Total integrated scatter (TIS) analysis

The Total Integrated Scatter (TIS) model was employed to estimate the specular (0°) reflectance of the measured nanostructured surfaces. In this framework, the reduction in coherent (specular) reflectance relative to that of an ideal flat surface is expressed in terms of the statistical properties of the surface height distribution. Specifically, under the Rayleigh–Rice perturbation theory, the mean specular reflectance can be approximated as $R_{spec} \approx R_{flat} \exp[-(4\pi\sigma \cos\theta_i/\lambda)^2]$, where $R_{flat}$ is the Fresnel reflectance of the corresponding smooth interface, $\sigma$ is the root-mean-square (RMS) surface height, $\theta_i$ is the incidence angle, and $\lambda$ is the wavelength in vacuum. This formulation assumes that the surface slopes are small and that multiple scattering and shadowing effects are negligible. For our samples, the RMS roughness is on the order of $\sigma \approx 7$–8 nm, so that $4\pi\sigma/\lambda_{min} \lesssim 0.3$ even at the shortest wavelength used ($\lambda_{min} = 350$ nm), placing the surfaces well within the validity range of first-order perturbation theory and the TIS model. Under these conditions, the coherent field attenuation is dominated by phase randomization rather than by multiple scattering, and the total scattered power can be accurately inferred from the surface height statistics alone. Compared with full-wave electromagnetic simulations such as finite-element (FEM) or finite-difference time-domain (FDTD) analysis, the TIS method provides a computationally efficient and physically transparent estimate of the specular reflectance, while remaining quantitatively reliable for shallow-relief, weakly scattering surfaces of the type considered here.

## Wettability test

For the artificial nanocoatings, a 3 μl water droplet was carefully placed on top of the sample surface. For the insect eyes, droplet positioning was done by a capillary attached to a micromanipulator, and registered by an inverted microscope (Axiovert 40 C).

The images were captured with a digital camera and analyzed with the Gwyddion software, measuring the droplet's contact angles from both sides.

## Nanostructures formation simulation

For the simulation, the software Ready, a cross-platform software for simulating reaction-diffusion systems, was used with two different scripts for in vivo and in vitro coating simulations. All parameters are listed in the Appendix Tables S2 and S5. The scripts for the in vivo and in vitro nanocoating simulations based on https://github.com/GollyGang/ready/tree/gh-pages/Patterns/Kryuchkov2020. For the 3D Lyapunov exponent plot Mathematica software was used. The script and exact parameters are provided in the Appendix.

## Protein purification

The NanoLuc luciferase sequence was amplified from the Go1-CASE plasmid (Larasati et al, 2022) with the following primers: forward, ctgttgactcgagccttacgccagaatgcgttcgcaca; reverse, tatgcagccaccacttctgtggagcgtggcctccgatccccgagtggttctcatccgcaacgaaaacctgtattttcagagcgtcttcacactcgaagattt. The reverse primer encompasses a linker with the sequence coding for the TEV protease cleavage site. The PCR product was subcloned with BstXI and XhoI restriction sites into the pET23Retinin plasmid (Kryuchkov et al, 2020), and the resulting plasmid was transformed into the BL21(DE3)pLysS *E. coli* strain for recombinant expression upon induction by 1 mM IPTG. The bacterial mass was lysed by the cell press. Retinin and RetLuc proteins were purified using the HisPur Ni-NTA resin following the manufacturer's recommendations, and further purified on the Superdex™ 200 Increase 10/300GL FPLC column using Tris-buffered saline (TBS: 20 mM Tris-HCl, and 150 mM NaCl, pH 7.6).

## Wave-guide interferometry

Grating Coupled Interferometry (GCI) experiments were conducted on a Creoptix WAVE delta system using 4PCH chips. The chips were conditioned with 100 mM sodium borate (pH 9.5) and 1 M NaCl. Retinin and RetLuc (100 µg/µl in 10 mM sodium acetate, pH 5.0) were immobilized on the chip surface using the standard amine coupling. This included 420 s of the surface activation with a 1:1 mix of 400 mM EDC and 100 mM NHS, 600 s Retinin or RetLuc injection, 420 s neutravidin (100 µg/µl) injection to fill the remaining sites, and a final 420 s surface passivation with 1 M ethanolamine at pH 8.0, yielding the surface masses of ca. 12,000 and 16,500 pg/mm² for Retinin and RetLuc, respectively. All preparation steps were performed at the 10 µl/min flow rate. Decanoic acid was injected in a 1:3 dilution series from 50 µM to 450 µM at 100 µl/min. Blank injections and a reference channel were used for double referencing. A 1:1 Langmuir binding model with bulk correction was used for all experiments.

## Wax emulsion preparation

4 g of carnauba wax was added to tubes with 40 ml 10% SDS solutions in water and sonicated in a water bath for 2 h at 80 °C. After subsequent 24 h incubation at room temperature (RT), the upper part enriched in wax nanodrops was diluted tenfold in 1xTBS

and further incubated for 48 h at RT. The upper part of the resulting mixture, enriched in wax drops bigger than 500 nm, was discarded. The lower part was dissolved roughly tenfold in 1xTBS to $OD_{600} = 0.5$. These emulsions are stable at RT for one year.

## Artificial nanocoatings

20 µl of a mixture of Retinin (1 mg/ml in TBS) or RetLuc (2 mg/ml in TBS) and the carnauba wax emulsion at proportions 3:1 and 3:2, respectively, were distributed evenly on a 1 cm² area of a glass coverslip and allowed to dry out gradually at different temperatures for 20 min at the humidity of 50–60%, rinsed in water, and re-dried. This process was repeated twice.

## Luminescence and protein amount measurements

Pieces of coverslip glass with the size of $4 \times 4$ mm were placed into a 96-well black microplate. Measurements were taken using the Infinite M Plex multifunctional plate reader with injections of 100 µl of 10 µM furimazine solution in TBS. Each piece of glass was removed, washed in water, and boiled for 1 h in the sample buffer (62.5 mM Tris-HCl, pH 6.8; 10% glycerol; 2% SDS; 1% β-mercaptoethanol; trace of bromophenol blue). These solutions were separated by 15% SDS-PAGE, stained by Coomassie and analyzed by using ImageJ.

## TIRF microscopy and diffusion coefficient quantification

TIRF microscopy was done at the Bioimaging core facility (University of Geneva, Switzerland) with the Nikon Eclipse Ti microscope, objective 100×1.49 Oil CFI Apochromat TIRF, and ORCA Fusion BT sCMOS camera. The same conditions were used as for the artificial nanocoating. Measurements were done at 25 °C. Retinin and RetLuc proteins were labeled by ATTO565 NHS-ester according to the manufacturer's protocol and used at the 1:20,000 ratio to the non-labeled proteins. BODIPY™ 500/510 $C_1$, $C_{12}$ (4,4-Difluoro-5-Methyl-4-Bora-3a,4a-Diaza-s-Indacene-3-Dodecanoic Acid) was diluted by carnauba wax emulsion 200,000-fold. Particle tracking and diffusion coefficient calculation were done by the MosaicSuite plugin for ImageJ.

## Statistical analysis

Statistical analysis was performed using GraphPad Prism software.

# Data availability

This study includes no data deposited in external repositories.

The source data of this paper are collected in the following database record: biostudies:S-SCDT-10_1038-S44319-025-00685-1.

# Peer review information

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

## Acknowledgements

We thank U. Steiner for kindly providing access to instruments at Adolphe Merkle Institute, B. Wilts for the help with optical measurements, P. Raia for help with wave-guide interferometry, and M. Savitsky and A. Koval for enlightening discussions. This work was funded by the Swiss National Science Foundation grant number 10.003.592 to VLK and the State research assignment 121032300105-0 at Lomonosov Moscow State University to VS. The authors acknowledge support from the European Cooperation in Science and Technology (COST) grant number CA21159.

## Author contributions

**Mikhail Kryuchkov**: Conceptualization; Formal analysis; Supervision; Investigation; Visualization; Writing—original draft; Project administration; Writing—review and editing. **Vladimir Savitsky**: Resources; Funding acquisition; Investigation; Writing—original draft. **Marc Jobin**: Resources; Formal analysis; Investigation; Writing—review and editing. **Stanislav Smirnov**: Software; Investigation; Methodology; Writing—review and editing. **Mirza Karamehmedović**: Software; Funding acquisition; Investigation; Methodology. **Jana Valnohova**: Investigation; Writing—review and editing. **Vladimir L Katanaev**: Resources; Supervision; Funding acquisition; Project administration; Writing—review and editing.

Source data underlying figure panels in this paper may have individual authorship assigned. Where available, figure panel/source data authorship is listed in the following database record: biostudies:S-SCDT-10_1038-S44319-025-00685-1.

## Disclosure and competing interests statement

The authors declare no competing interests.

# Expanded View Figures

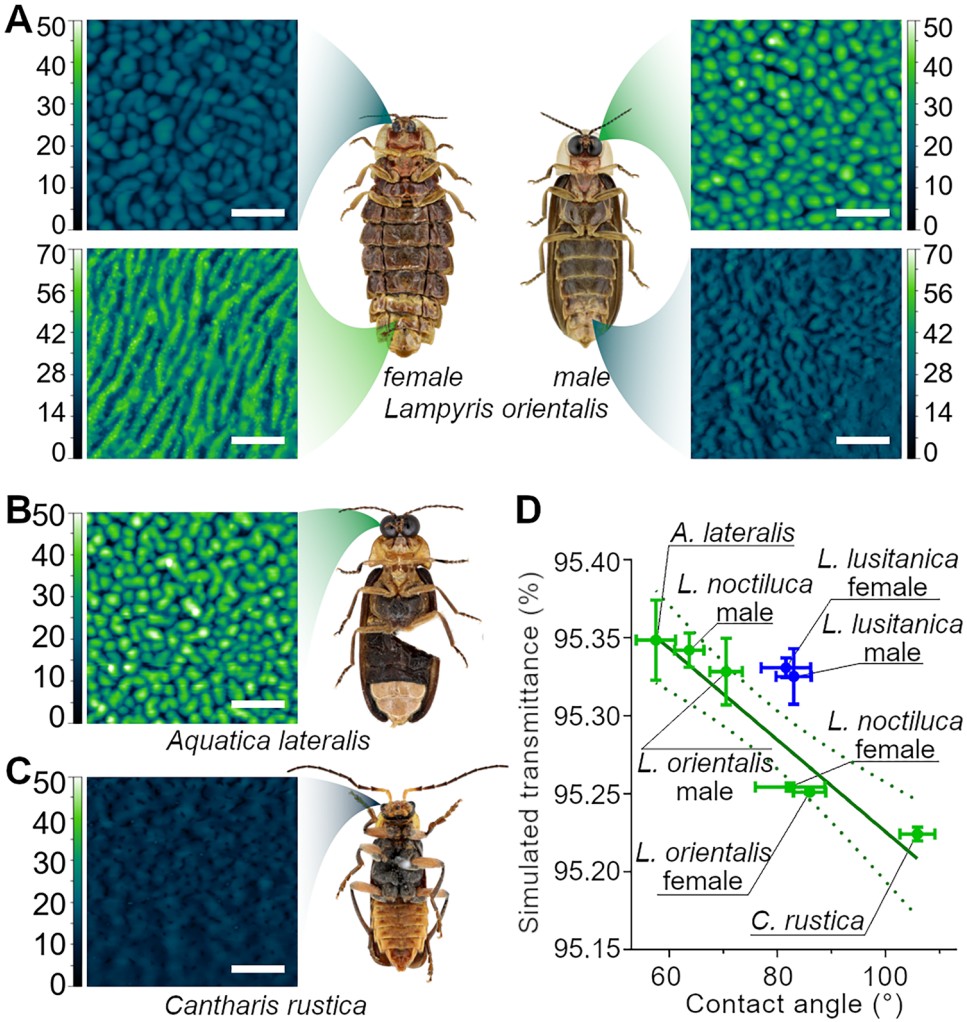

**Figure EV1.  Comparison of surface nanostructures and their optical-wetting properties in different beetle species.**

(**A–C**) Light microscopy images and AFM height maps of (**A**) *L. orientalis* corneal and lantern nanocoatings, showing structural dimensions and organization closely resembling those of *L. noctiluca*; (**B**) *A. lateralis*; and (**C**) *C. rustica* corneal nanocoatings, illustrating interspecific variation in nanoscale organization. (**D**) Correlation between measured contact angle and simulated optical transmittance at $\lambda = 550$ nm. Species with more prominent nanocoatings exhibit higher wettability and enhanced optical transmission, following an approximately linear trend except for *L. lusitanica* (blue). Lines represent linear regression with 99% confidence intervals. For wettability tests, $n = 8$ and $N = 2$ (biologically independent animals); only for *A. lateralis*, $N = 1$. Simulations were performed using AFM scans from two biologically independent animals ($n = 2$, $N = 2$), except *A. lateralis* ($N = 1$). Scale bars: 1 μm. Source data are available online for this figure.

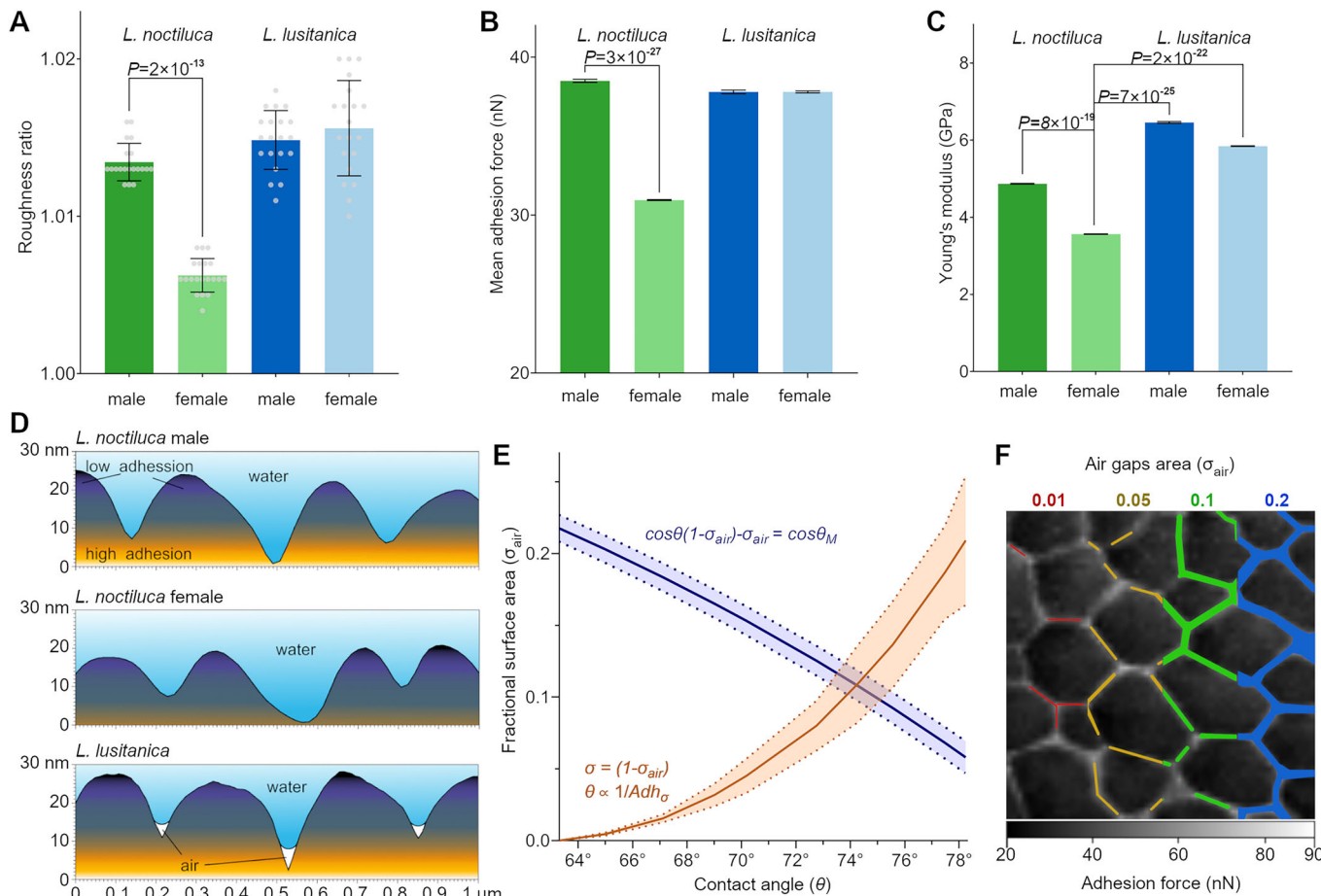

**Figure EV2.  Wenzel and Cassie-Baxter mechanisms explain species-specific anti-wettability of firefly corneal nanocoatings.**

(A) In *L. noctiluca*, the difference in hydrophobicity between males and females cannot be explained by surface topography alone. According to the Wenzel relation ($\cos \Theta_M = r \times \cos \Theta_Y$), the observed contact angles would require an unrealistically high roughness ratio ($r_{male}/r_{female} = 6.3$), whereas AFM measurements show nearly identical roughness ($r_{male} = 1.013$, $r_{female} = 1.006$). Data are shown as mean ± SD; $n = 20$. (B, C) AFM adhesion and elasticity maps reveal that female corneae are less adhesive and more elastic, consistent with a higher wax-to-protein ratio in their nanocoatings (Anderson and Gaimari, 2003; Kryuchkov et al, 2020; Nickerl et al, 2014). Thus, *L. noctiluca* follows the Wenzel model, where surface chemistry rather than roughness governs wettability. Data are mean ± SEM; $n = 25{,}000$–$60{,}000$. (D) In *L. lusitanica*, tightly packed, paving-stone-like nanostructures create air pockets, consistent with the Cassie-Baxter model ($\cos \Theta \times (1 - \sigma_{air}) - \sigma_{air} = \cos\Theta_M$ (Bello et al, 2023; Kryuchkov et al, 2017a)). (E) To quantify the air-covered fraction ($\sigma_{air}$), we combined the Cassie-Baxter equation (blue curve) with an empirical correlation between contact angle and adhesion derived from *L. noctiluca* ($\Theta = 162.4 - 2.45 \times Adh$ (Drelich, 2019); orange curve). The two curves intersect at $\sigma_{air} \approx 0.11$, indicating that 11% of the *L. lusitanica* surface is occupied by trapped air. (F) Adhesion force maps were segmented to exclude the grooves (air-trapping zones) between nanostructures, producing the $Adh_\sigma$ data used in (E). Schema illustrates increasing $\sigma_{air}$ from 0.01 to 0.2. Adhesion force measurement scan ($1 \times 1\,\mu m$) is shown as background. Statistical significance in (A–C) is determined with t-test. Together, these analyses show that *L. noctiluca* hydrophobicity arises from chemical composition (Wenzel regime), whereas *L. lusitanica* achieves superior water repellency through nanoscale air entrapment (Cassie-Baxter regime).

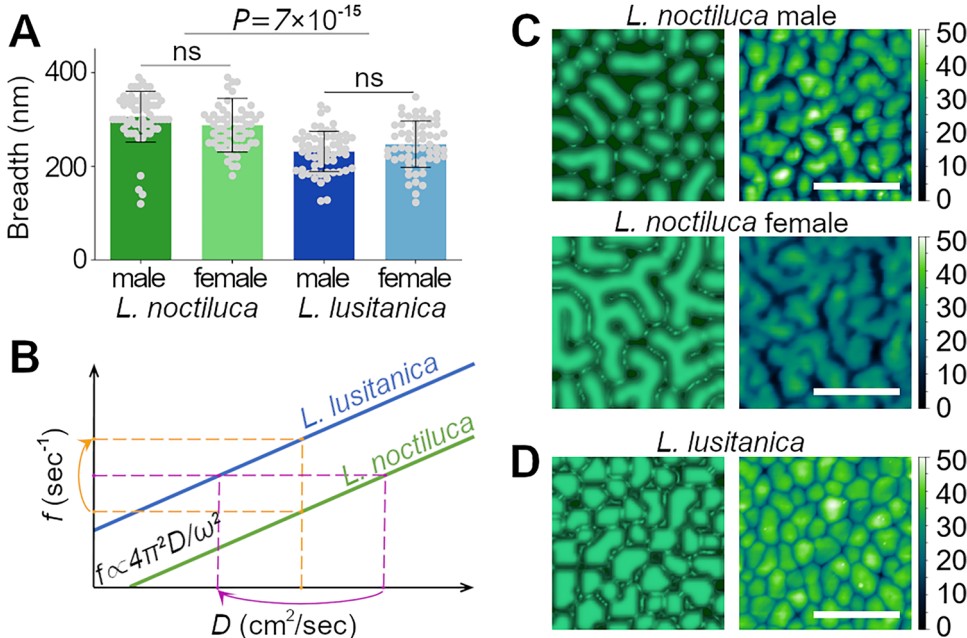

**Figure EV3. Firefly nanostructures' comparative breadth as the entry point to set the Turing modeling parameters.**

(A) The size of the paving-stones-like structures is significantly narrower than that of the nipples-like and maze-like structures, hinting at a lower activator diffusion parameter. Data are shown as mean ± SD, $n = 60$, $N = 2$ (biologically independent animals), "ns" indicates the absence of significant difference, t-test. (B) The relationship between nanostructure size ($\omega$), reaction rate ($f$), and diffusion coefficient ($D$) indicates that the transition from *L. noctiluca* to *L. lusitanica* nanostructures is possible only through an increase in $f$ (orange) or a decrease in $D$ (pink). (C) Turing modeling (left panels) efficiently recapitulates experimentally observed (right panels) *L. noctiluca* male (nipples) and female (maze-like) corneal nanocoatings. For the AFM scans, surface height is indicated by the color scale shown next to the images. (D) Similarly, the *L. lusitanica* paving-stones-like pattern (experimental image on the right) is faithfully recapitulated by simulation (on the left) using the activator diffusion coefficient twice as low as that used to model *L. noctiluca* structures. The exact parameters applied in the simulations in (C, D) are provided in Appendix Table S2. Scale bars: 1 µm, height is indicated in nm. Source data are available online for this figure.

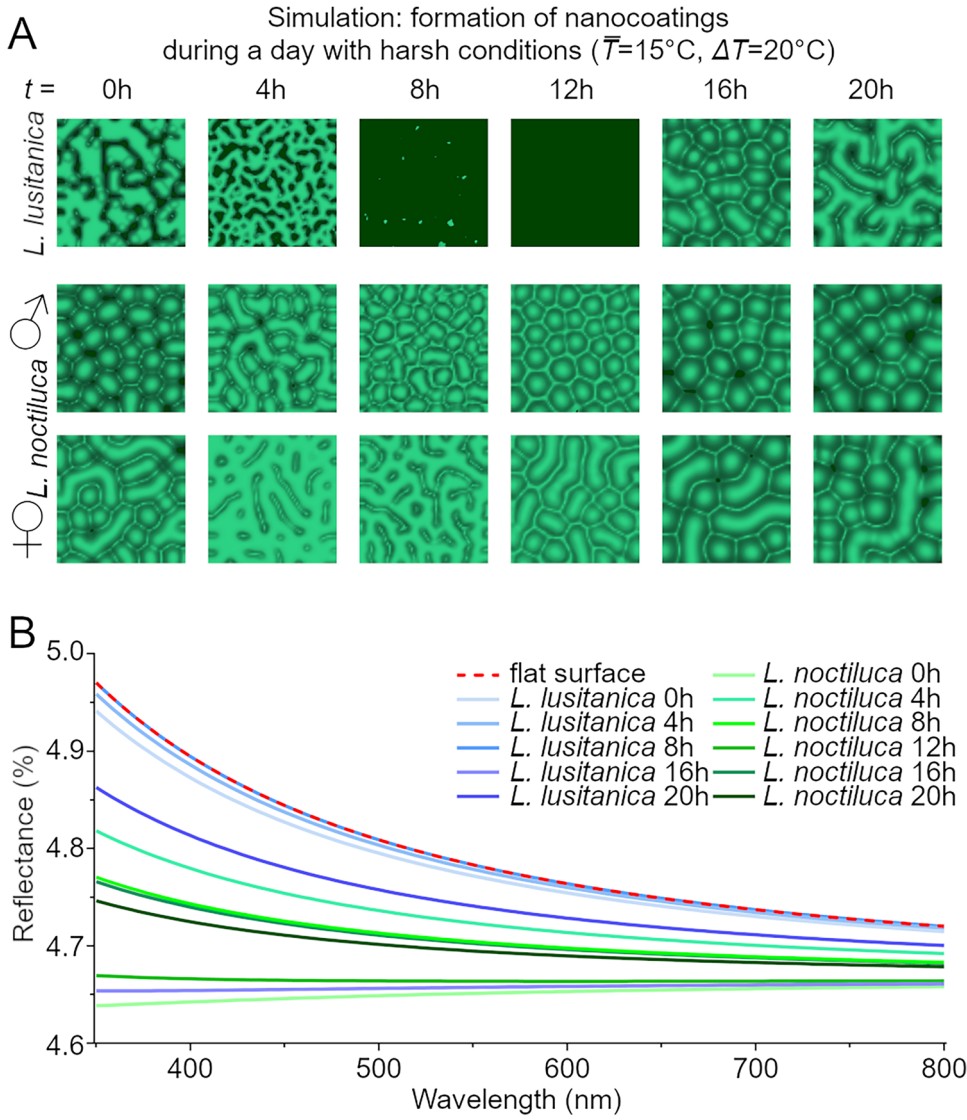

**Figure EV4. Differential thermal resilience of nanocoating formation in *L. lusitanica* and *L. noctiluca*.**

(A) Simulations illustrate that the nanocoating patterns of *L. lusitanica* are highly sensitive to diurnal thermal fluctuations, whereas those of *L. noctiluca* remain stable. Each row shows the progression of surface patterning at different time points ($t = 0$–$20$ h) during a day characterized by a mean temperature of 15 °C and an amplitude of ±10 °C ($\Delta T = 20$ °C; red dot in Fig. 3D). Periods of pattern disappearance reflect a complete loss of Turing instability. Simulation parameters are described in Appendix Note 4. (B) Simulated optical reflectance spectra demonstrate a marked loss of functional surface properties for *L. lusitanica* nanocoatings (blue curves), in contrast to the more stable optical response of *L. noctiluca* (green curves). The dashed red line indicates a flat, patternless reference surface. Source data are available online for this figure.

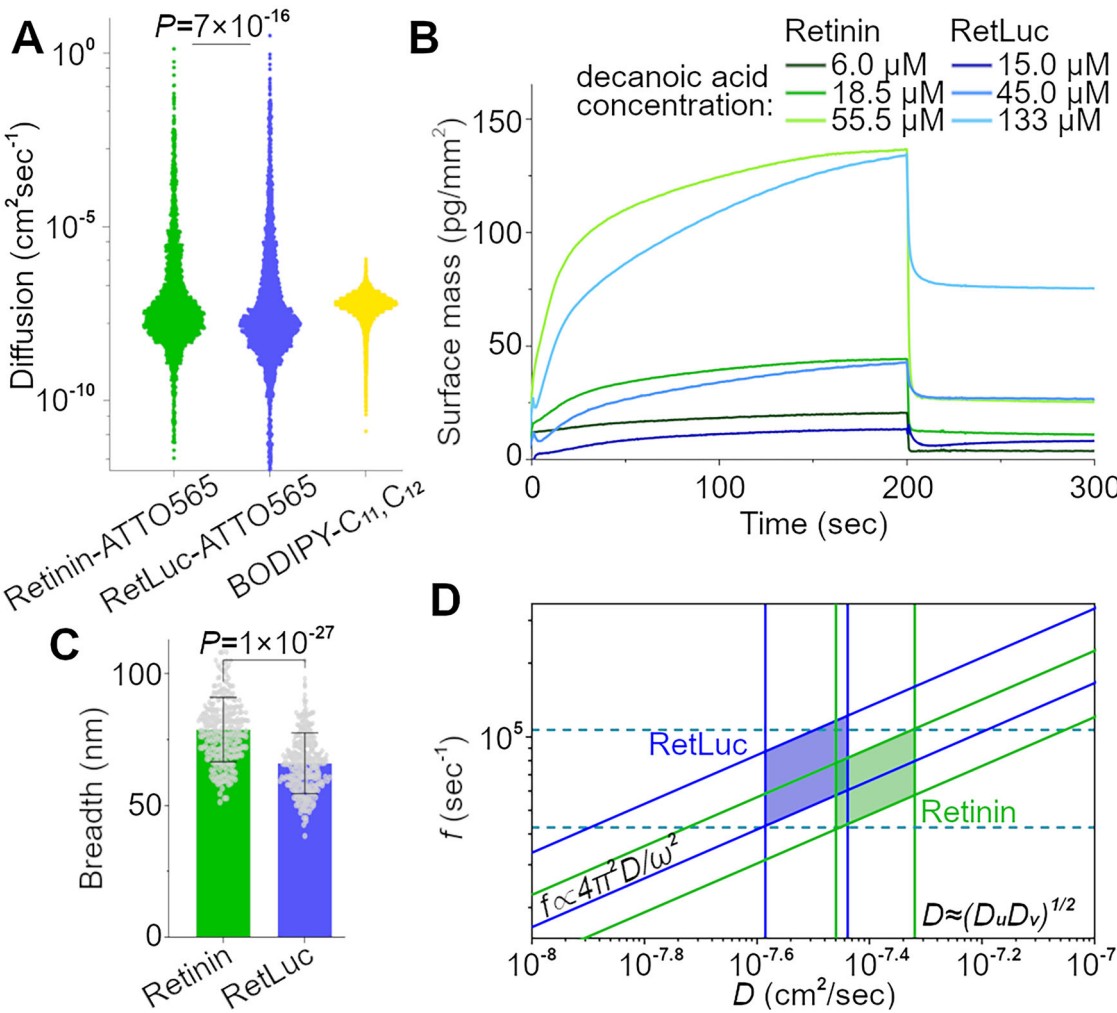

**Figure EV5. Biophysical characterization of Retinin-NanoLuc fusion protein confirms preserved reaction kinetics and reduced molecular mobility.**

(A) Diffusion coefficient, measured by individual molecules tracking method, decreases in the case of Retinin and RetLuc, compared to native Retinin. $n = 6723$ (Retinin) and $n = 6455$ (RetLuc), $N = 3$ (independent experiments); in the case of the inhibitor (BODIPY 500/510 C1,C12) $n = 9988$, with $N = 5$ (independent experiments). (B, C) Wave-guide interferometry of the interaction between Retinin (B) or RetLuc (C) with decanoic acid demonstrates almost identical properties of these proteins. (C) The size of the RetLuc-based structures is significantly smaller than Retinin-based ones. Data are shown as mean ± SD, $n = 250$, $N = 3$ (independent experiments). (D) The parallelograms defined by the overlap of the joint diffusion coefficient (D) and the reaction rate constant (f) areas, shown as mean ± SD, represent the zones within the parameter space where the experimentally observed nanocoatings are allowed to form through the reaction-diffusion mechanism. Note that the two zones reside within the range area of $f$ (dashed turquoise lines), serving as indirect proof that the fusion of Retinin with NanoLuc does not alter the reaction rate constant of the protein.

