## [Peer Review File · EMBO Reports]

Trade-offs in insect eye nanocoatings: Implications for vision, ecology, and climate sensitivity

Mikhail Kryuchkov, Vladimir Savitsky, Marc Jobin, Stanislav Smirnov, Mirza Karamehmedović, Jana Valnohova, and Vladimir Katanaev

Corresponding author(s): Mikhail Kryuchkov (mikhail.kryuchkov@unige.ch) , Vladimir Katanaev (vladimir.katanaev@unige.ch),

Review Timeline:

Submission Date:	16th May 25
Editorial Decision:	13th Aug 25
Revision Received:	12th Nov 25
Editorial Decision:	7th Dec 25
Revision Received:	9th Dec 25
Accepted:	15th Dec 25

Editor: Yehu Moran

Transaction Report:

Dear Dr Kryuchkov

Thank you for the submission of your manuscript to EMBO Reports.

I would like to start by apologizing once again for the extremely long time it took us to get back to you with a decision letter. This is due to the highly complex nature of your interesting manuscript that is highly interdisciplinary. It was important for us to receive feedback also from an expert on the ecological and evolutionary side of things and I am happy to report we finally succeeded and as you will see this expert was positive about your paper.

We have now received the full set of referee reports that are all pasted below.

In general all three are positive but also have comments that require your attention.

I would thus like to invite you to revise your manuscript with the understanding that the referee concerns must be fully addressed and their suggestions taken on board. Please address all referee concerns in a complete point-by-point response. Acceptance of the manuscript will depend on a positive outcome of a second round of review. It is EMBO Reports policy to allow a single round of major revision only and acceptance or rejection of the manuscript will therefore depend on the completeness of your responses included in the next, final version of the manuscript.

We realize that it is difficult to revise to a specific deadline. In the interest of protecting the conceptual advance provided by the work, we recommend a revision within 3 months (13th Nov 2025). Please discuss the revision progress ahead of this time with the editor if you require more time to complete the revisions.

- 1) A data availability section providing access to data deposited in public databases is missing. If you have not deposited any data, please add a sentence to the data availability section that explains that.
- 2) Your manuscript contains statistics and error bars based on $n=2$. Please use scatter blots in these cases. No statistics should be calculated if $n=2$.

3) We replaced Supplementary Information with Expanded View (EV) Figures and Tables that are collapsible/expandable online. A maximum of 5 EV Figures can be typeset. EV Figures should be cited as 'Figure EV1, Figure EV2' etc... in the text and their respective legends should be included in the main text after the legends of regular figures.

5) a complete author checklist, which you can download from our author guidelines

<<https://www.embopress.org/page/journal/14693178/authorguide>>. Please insert information in the checklist that is also reflected in the manuscript. The completed author checklist will also be part of the RPF.

6) Please note that all corresponding authors are required to supply an ORCID ID for their name upon submission of a revised manuscript (<<https://orcid.org/>>). Please find instructions on how to link your ORCID ID to your account in our manuscript tracking system in our Author guidelines

<<https://www.embopress.org/page/journal/14693178/authorguide#authorshipguidelines>>

10) Regarding data quantification (see Figure Legends:

<https://www.embopress.org/page/journal/14693178/authorguide#figureformat>)

- the name of the statistical test used to generate error bars and P values,

- the number (n) of independent experiments (please specify technical or biological replicates) underlying each data point,

- the nature of the bars and error bars (s.d., s.e.m.),

- If the data are obtained from n Program fragment delivered error ``Can't locate object method "less" via package "than" (perhaps you forgot to load "than"?) at //ejpvfs23/sites23b/embor_www/letters/embor_decision_revise_and_review.txt line 56.' 2, use scatter blots showing the individual data points.

12) All Materials and Methods need to be described in the main text using our 'Structured Methods' format, which is required for all research articles. According to this format, the Methods section includes a Reagents and Tools Table (listing key reagents, experimental models, software and relevant equipment and including their sources and relevant identifiers) followed by a Methods and Protocols section describing the methods using a step-by-step protocol format. The aim is to facilitate adoption of the methodologies across labs. More information on how to adhere to this format as well as a downloadable template (.docx) for the Reagents and Tools Table can be found in our author guidelines:

An example of a Method paper with Structured Methods can be found here: <https://www.embopress.org/doi/full/10.1038/s44320-024-00037-6#sec-4>

I look forward to seeing a revised form of your manuscript when it is ready.

Yours sincerely,

Yehu Moran
Academic Editor
EMBO Reports

Referee #1:

This paper studies the physical properties of the anti-reflective structures of firefly lantern as well as eyes. Extensive chemical experiments and reaction-diffusion modeling revealed subtle dependencies of the surfaces' creation on temperature. This study fits well in a long list of similar reports on nanostructured surfaces of various insects by the same team. The claim of the relevance for climate studies and evolutionary trajectories may be a bit far-fetched, as the optical and adhesive effects are not extreme, but the report is detailed and scientifically sound, and thus valuable. The reported ratio of the different reflection spectra may be OK, but this is tricky, because the spatial distribution of the reflected light may distinctly vary when the reflecting surface structures differ. Strictly speaking, comparing reflectance amplitudes can therefore only be reliably done when measuring with an integrating sphere. But, when the surface structure is known, it would have been a logical step to model the spatial reflectance characteristics, so to ascertain whether the implicitly assumed specularity of the surfaces is justified. FDTD modeling thus might have been applied. The references need a closer look. For instance,

- Ref. 1 is incorrect. The book is from 1999 and attributed to Bass et al.
- Ref. 2 is incomplete, Page (11) is missing.
- Ref. 4 seems to be identical to Ref. 9.
- Ref. 5: Cham???
- Ref. 16: 10, 163
- Ref. 17 and Ref. 55 are also identical
- Ref. 20: ACS Appl. Bio Mater. 2025, 8, 1, 784-791
- Ref. 23: 1388
- Ref. 33 ... and so on.

Referee #3:

Kryuchkov et al. investigate the structural and functional properties of corneal nanocoatings in *Luciola lusitanica* and *Lampyrus noctiluca* fireflies. The authors report the presence of a unique "paving-stones-like" nanocoating that combines anti-reflective and hydrophobic properties and explore its temperature sensitivity through both Turing modeling and synthetic assembly. The manuscript is comprehensive and can be well followed. However, I have several concerns and suggestions for the authors to address, which I hope will improve the clarity and scientific impact of the work.

The present work is, to a large degree, an extension of the authors' previous studies (e.g., Kryuchkov et al. 2020, 2021) on nanocoating patterning in insect eyes. While the observation of dual-function nanostructures in *L. lusitanica* is novel, the approach, modeling framework, and much of the narrative closely parallel earlier publications from the same group. Please clarify what is fundamentally new here, beyond application to a new species.

The key physical characterization (Figure 2A) compares reflectance across two distantly related species. Is it valid to benchmark "reflectance efficiency" of *L. lusitanica* against *L. noctiluca*? These species differ in ecology, behavior, and likely cuticle chemistry. Could the authors justify this comparison more fully, and ideally include an additional reference species, or discuss possible confounding factors (e.g., habitat, light environment, evolutionary distance)? What about intra-species or population-

level variability? Why not refer against a normal transmission standard?

The discussion makes strong claims regarding visual performance, signaling efficiency, and evolutionary advantage, but these are not directly demonstrated. There are no behavioral experiments or ecological observations linking nanostructure morphology to actual fitness outcomes (e.g., mate choice, predation, or habitat occupancy). Could the authors support these statements with additional data or literature?

The reaction-diffusion modeling is elegant (but was shown before, the figure is a bit hard to read!), but its biological relevance is not fully demonstrated. There is no modeling of optical performance under natural conditions, nor of how temperature-driven nanostructure changes would affect the animal's signaling or vision. Please consider quantitative analysis or at least discuss the limitations of inferring ecological consequences solely from in vitro measurements.

The claim that nanostructure assembly is highly temperature-sensitive and evolutionarily limiting is central to the manuscript, yet seems supported by very limited in vivo data (e.g., one or two high-altitude specimens, Figure 5B/C). Have similar levels of structural variability been demonstrated in wild insect populations or in response to natural temperature variation? The claim that a 0.2 deg C mean increase could cause "widespread developmental failures" appears highly speculative.

Minor Points:

The explanation of the anti-reflectivity vs. hydrophobicity "trade-off" could be clarified, as some recent work suggests this may not always be mutually exclusive.

Please ensure all axes and units are fully labeled. Add error bars where appropriate.

Referee #4:

Kruchkov et al. describe an interesting phenomenon regarding the abdominal cuticle and the eye coating in fireflies. They show that both cuticles are optimized to allow light to pass through the cuticle, at the expense of keeping the cuticle free of fouling.

They demonstrate this in two species of fireflies, and show that different temperature ranges of the species as well differences in mating ecology, lead to a difference in the distribution of the light transmitting cuticle.

I was asked to give my assessment on the evolutionary and behavioral/ecological aspects of the manuscript, since I am not an expert in material science or in cuticle structure. I find the ecological and evolutionary story to be very interesting and worthy of publication. The authors provide a beautiful example of an adaptation manifested in the micro-structure scale. I am not aware of many similar examples that demonstrate a link between a specific selective pressure and a rapidly varying structure with nanoscale properties. The link is of course speculative, but the evidence the authors provide is well supported. Based on the organismic narrative, without assessing the technical aspects of the paper, I think the paper is worthy of publication and suitable for EMBO reports.

Dear Editors,

We are pleased to submit the revised version of our manuscript entitled

"Trade-offs in insect eye nanocoatings: Implications for vision, ecology, and climate sensitivity" for further consideration at EMBO Reports.

We are grateful to the reviewers for their constructive and insightful comments, which have greatly improved the clarity and impact of our work. In this revised version, we have carefully addressed all points raised:

- We expanded the comparative dataset to include three additional species, providing a broader ecological and evolutionary context (Figure EV1).
- We clarified the novelty of our study by emphasizing that it introduces the first thermodynamic framework linking nanoscale organization, environmental sensitivity, and evolutionary adaptation.
- We improved figure clarity and readability, including the addition of numerical axis labels and verification of all error bars (e.g., Figure 3C).
- We added new simulations illustrating how temperature variations affect optical performance (Figure EV4), providing quantitative support for the thermosensitive behavior of nanocoatings.
- We expanded and refined the Results and Discussions sections (pp. 3–9) to clarify the interpretation of ecological and visual implications, include population-level variability, and provide additional examples from related insect taxa.

We believe these revisions have substantially strengthened the manuscript, reinforcing both its mechanistic rigor and evolutionary scope. The study now presents a clearer and more comprehensive picture of how thermosensitive nanocoatings contribute to visual adaptation and may constrain species' resilience under changing climatic conditions.

We thank the reviewers and the editorial team for their thoughtful evaluation and helpful feedback, and we hope that this revised version meets the journal's standards for publication.

Sincerely,

Mikhail Kryuchkov, PhD

University of Geneva

mikhail.kryuchkov@unige.ch

Referee #1:

- This paper studies the physical properties of the anti-reflective structures of firefly lantern as well as eyes. Extensive chemical experiments and reaction-diffusion modeling revealed subtle dependencies of the surfaces' creation on temperature.

This study fits well in a long list of similar reports on nanostructured surfaces of various insects by the same team. The claim of the relevance for climate studies and evolutionary trajectories may be a bit far-fetched, as the optical and adhesive effects are not extreme, but the report is detailed and scientifically sound, and thus valuable.

- *We thank the reviewer for this thoughtful comment. While we agree that the observed optical and adhesive effects are modest, we intended to highlight that even such subtle nanoscale features may have important developmental and ecological implications. Our reaction-diffusion simulations predict that further temperature changes could destabilize nanocoating assembly entirely. Since nanocoatings form prior to other cuticular layers, such disruption may impair cuticularization and potentially lead to non-viable phenotypes. Thus, nanoscale adaptations that enhance optical or self-cleaning performance can simultaneously impose tight environmental constraints on development.*

- The reported ratio of the different reflection spectra may be OK, but this is tricky, because the spatial distribution of the reflected light may distinctly vary when the reflecting surface structures differ. Strictly speaking, comparing reflectance amplitudes can therefore only be reliably done when measuring with an integrating sphere. But, when the surface structure is known, it would have been a logical step to model the spatial reflectance characteristics, so to ascertain whether the implicitly assumed specularity of the surfaces is justified. FDTD modeling thus might have been applied.

- *Indeed, we have performed simulations using the Total Integrated Scattering (TIS) approach, which is well-suited for the range of surface roughness observed in our samples (Figure EV1, EV4). We did not apply full FDTD simulations in this work due to computational limitations. However, in comparable cases where we performed FDTD analyses, the results were in good agreement with those obtained via TIS, supporting the reliability of our approach. Notably, while the trend of reflectance differences was consistent across simulations and experiments, the overall reflectance efficiency predicted by TIS was lower than in real measurements, likely reflecting the effects of complex surface curvature, as the reviewer rightly pointed out, or possibly the internal dipole organization within the nanostructures.*

- The references need a closer look. For instance,
 - Ref. 1 is incorrect. The book is from 1999 and attributed to Bass et al.
 - Ref. 2 is incomplete, Page (11) is missing.
 - Ref. 4 seems to be identical to Ref. 9.
 - Ref. 5: Cham???
 - Ref. 16: 10, 163
 - Ref. 17 and Ref. 55 are also identical
 - Ref. 20: ACS Appl. Bio Mater. 2025, 8, 1, 784-791
 - Ref. 23: 1388
 - Ref. 33 ... and so on.

- *We carefully reviewed and corrected all references as suggested. The inaccuracies and duplications mentioned have been verified and fixed accordingly.*

Referee #3:

Kryuchkov et al. investigate the structural and functional properties of corneal nanocoatings in *Luciola lusitanica* and *Lampyris noctiluca* fireflies. The authors report the presence of a unique "paving-stones-like" nanocoating that combines anti-reflective and hydrophobic properties and explore its temperature sensitivity through both Turing modeling and synthetic assembly. The manuscript is comprehensive and can be well followed. However, I have several concerns and suggestions for the authors to address, which I hope will improve the clarity and scientific impact of the work.

- The present work is, to a large degree, an extension of the authors' previous studies (e.g., Kryuchkov et al. 2020, 2021) on nanocoating patterning in insect eyes. While the observation of dual-function nanostructures in *L. lusitanica* is novel, the approach, modeling framework, and much of the narrative closely parallel earlier publications from the same group. Please clarify what is fundamentally new here, beyond application to a new species.

- *We thank the reviewer for this comment and the opportunity to clarify the novelty of our work. While our previous publications established the general principles of reaction-diffusion-driven nanocoating formation, the present study goes substantially beyond these earlier findings. Here, for the first time, we investigate the thermodynamic behavior of reaction-diffusion nanocoatings, showing that their morphology is not fixed but dynamically responds to temperature. This thermodynamic perspective is essential, as it anchors the RD framework in physical reality, connecting the pattern-forming mechanism to measurable energetic and environmental parameters. Moreover, we link this thermodynamic adaptability to evolutionary and ecological context, showing how the nanoscale organization in *Luciola**

lusitanica and *Lampyris noctiluca* correlates with their habitats and light-emission-perception strategies (p.9). Thus, the present study establishes the first mechanistic and evolutionary connection between thermodynamic regulation, nanocoating structure, and ecological adaptation: transforming the RD framework from an abstract model into a physically and biologically grounded concept.

- The key physical characterization (Figure 2A) compares reflectance across two distantly related species. Is it valid to benchmark "reflectance efficiency" of *L. lusitanica* against *L. noctiluca*? These species differ in ecology, behavior, and likely cuticle chemistry. Could the authors justify this comparison more fully, and ideally include an additional reference species, or discuss possible confounding factors (e.g., habitat, light environment, evolutionary distance)?

- We agree that comparing *L. lusitanica* and *L. noctiluca* alone could be confounded by ecological and phylogenetic differences. To address this, we expanded the analysis to include three additional reference species representing distinct ecological and evolutionary contexts (pp.3-4, Fig. EV1). This broader comparison confirms that reflectance and self-cleaning properties follow a near-linear trade-off across species, consistent with shared functional constraints rather than species-specific chemistry or habitat effects. Moreover, by incorporating these data, we were able to discuss the potential influence of surface curvature and nanoscale molecular organization on optical efficiency, thereby contextualizing interspecies variation as the reviewer suggested.

- What about intra-species or population-level variability?
 - To assess potential intra-species and population-level variability, we analyzed multiple individuals from different localities and years (clarified in the Methods). Within each species, we found only minor variations in nanostructure organization and reflectance characteristics. The only notable deviation was observed in a single *Luciola lusitanica* specimen collected at 1950 m altitude. Overall, these results suggest that the described nanocoating traits are consistent within species.

- Why not refer against a normal transmission standard?
 - We performed the measurements in a relative manner, as referencing against a flat transmission standard would yield physically meaningless results for strongly curved corneal surfaces. Due to the complex and uneven curvature of insect eyes, such a comparison would not accurately represent the actual optical behavior of the nanostructured cuticle. Our relative approach therefore provides a more realistic basis for comparing reflectance between species under identical measurement conditions.

- The discussion makes strong claims regarding visual performance, signaling efficiency, and evolutionary advantage, but these are not directly demonstrated. There are no behavioral experiments or ecological observations linking nanostructure morphology to actual fitness outcomes (e.g., mate choice, predation, or habitat occupancy). Could the authors support these statements with additional data or literature?

- *We thank the reviewer for this insightful comment. We fully agree that direct behavioral or ecological experiments linking nanostructural morphology to fitness outcomes remain limited. In response, we have revised the Introduction (pp. 2-3) to better delineate the scope of our claims and to incorporate recent evidence that establishes indirect but compelling links between nanoscale morphology and organismal performance. Specifically, we now refer to *Drosophila melanogaster*, where downregulation of *Retinin* shortens corneal nanocoatings and leads to impaired phototaxis and reduced lifespan, and domesticated *Bombyx mori*, which has lost functional corneal nanocoatings relative to its wild ancestor *B. mandarina*, consistent with relaxed selection under laboratory rearing. We have also added comparative examples from other taxa, such as Gyrinidae beetles and owlflies, demonstrating ecological specialization of corneal nanocoatings.*

- The reaction-diffusion modeling is elegant (but was shown before, the figure is a bit hard to read!)

- *To improve clarity, we have modified Figure 3 by splitting it, and added Figure EV4 to provide additional detail. We believe these changes make the reaction-diffusion modeling results much easier to read while preserving the key insights.*

- , but its biological relevance is not fully demonstrated. There is no modeling of optical performance under natural conditions, nor of how temperature-driven nanostructure changes would affect the animal's signaling or vision. Please consider quantitative analysis or at least discuss the limitations of inferring ecological consequences solely from *in vitro* measurements.

- *We have now added simulations demonstrating that *L. lusitanica* nanocoatings lose functional optical properties under temperature changes, whereas *L. noctiluca* remains more stable (Figure EV4, p. 4). This provides a quantitative illustration of potential biological consequences, while we acknowledge that further *in vivo* studies would be needed to fully establish ecological relevance.*

- The claim that nanostructure assembly is highly temperature-sensitive and evolutionarily limiting is central to the manuscript, yet seems supported by very limited *in vivo* data (e.g., one or two high-altitude specimens, Figure 5B/C). Have similar levels of

structural variability been demonstrated in wild insect populations or in response to natural temperature variation?

- *Indeed, we were able to identify only a single specimen showing clear alterations in nanocoating morphology. This rarity may reflect both the potentially deleterious nature of such deviations, which could reduce fitness and thus be evolutionarily eliminated, and the exceptionally tight developmental control governing nanocoating formation. While available data on natural structural variability remain scarce, we have now expanded the Results and Discussion to include all known examples of temperature, or environment-induced nanostructural changes reported in arthropods (pp. 6-7). These additions help place our observations in a broader biological context and further underscore how finely regulated and evolutionarily constrained such nanostructures appear to be.*

- The claim that a 0.2 deg C mean increase could cause "widespread developmental failures" appears highly speculative.

- *The statement has been revised to clarify that our prediction refers to a threshold-driven instability in the nanocoating formation process rather than a deterministic global outcome. We now explicitly relate this modeled sensitivity to empirically documented threshold effects in other biological systems, where temperature shifts as small as 0.1–0.3 °C have been shown to produce disproportionate developmental or phenological impacts (p. 9). These studies collectively demonstrate that minor mean temperature increases can advance spring phenology, increase exposure to frost events, and lead to tissue damage or developmental failure in sensitive taxa. We believe this revision makes the statement more cautious, mechanistically grounded, and empirically supported, addressing the reviewer's concern while preserving the conceptual link between temperature sensitivity and developmental instability.*

- The explanation of the anti-reflectivity vs. hydrophobicity "trade-off" could be clarified, as some recent work suggests this may not always be mutually exclusive.

- *We appreciate this valuable comment. Indeed, we initially referred to this as a "rare" or atypical trade-off, and we now clarify that anti-reflective and hydrophobic functions are not strictly mutually exclusive. In the revised the Results and Discussion, we have included some examples of systems where both properties coexist, along with an explanation of the physical and developmental constraints that limit such dual functionality (p. 4). Specifically, we note that achieving both effects simultaneously may lead to the oscillatory mode of nanostructure formation, which restrict the parameter space for stable pattern emergence. These additions help contextualize our findings within the broader spectrum of known nanostructural multifunctionality.*

- Please ensure all axes and units are fully labeled. Add error bars where appropriate.

- *We added numerical labels to the axes in Figure 3C and verified that all figures include appropriate error bars.*

Referee #4:

Kruchkov et al. describe an interesting phenomenon regarding the abdominal cuticle and the eye coating in fireflies. They show that both cuticles are optimized to allow light to pass through the cuticle, at the expense of keeping the cuticle free of fouling. They demonstrate this in two species of fireflies, and show that different temperature ranges of the species as well as differences in mating ecology, lead to a difference in the distribution of the light transmitting cuticle.

I was asked to give my assessment on the evolutionary and behavioral/ecological aspects of the manuscript, since I am not an expert in material science or in cuticle structure. I find the ecological and evolutionary story to be very interesting and worthy of publication. The authors provide a beautiful example of an adaptation manifested in the micro-structure scale. I am not aware of many similar examples that demonstrate a link between a specific selective pressure and a rapidly varying structure with nanoscale properties. The link is of course speculative, but the evidence the authors provide is well supported. Based on the organismic narrative, without assessing the technical aspects of the paper, I think the paper is worthy of publication and suitable for EMBO reports.

- *We sincerely thank the reviewer for their thoughtful and encouraging assessment. We are glad that the ecological and evolutionary aspects of our study, as well as the organismic narrative linking selective pressures to nanoscale structural adaptations, were found interesting and compelling. We also believe that the inclusion of three additional species as controls has further strengthened the manuscript, providing a broader comparative context and reinforcing our conclusions. Your positive assessment and endorsement of the manuscript for EMBO Reports is greatly appreciated.*

Dear Dr. Kryuchkov

Thank you for the submission of your revised manuscript to our offices. We have now received the enclosed reports from the referees that were asked to assess it. EMBOR-2025-61953V2 still has minor suggestions that I would like you to incorporate before we can proceed with the official acceptance of your manuscript. Please ignore the not so nice comments from Referee #3. We do not share their opinion regarding the interest level of your paper (otherwise we would not continue this revision process).

I also enclose below comments from our editorial assistance team that require your attention before we can formally accept your manuscript.

I look forward to seeing a new revised version of your manuscript as soon as possible.

Yehu Moran
Academic Editor
EMBO Reports

Comments from editorial assistance team

Conflict of Interest: included, but it needs to be renamed to Disclosure and Competing Interests Statement

AC/CRedit: needs to be removed from the manuscript text and included only in our system.

REFERENCES: if a reference has more than 10 authors, et al should be used after the 10th name. Also, please see comments for Referee #1

DATA NOT SHOWN: OK

FUNDING INFO: OK, but the following grant should also be entered in the system in the grant field: E-COST-GRANT-CA21159-a405c60a.

FIGURE CALLOUTS: missing callouts for - individual panel of Figure 1 A-D, Figure 4C, Appendix Table S5; "supplementary" nomenclature should not be used. Please correct.

DATASET EV LEGENDS:

APPENDIX FILE WITH Table of Content: in, but the Table of Content should list each item and its title page; the nomenclature is not OK, needs to be Appendix Table S1, etc, Appendix Figure S1, etc. throughout the Appendix file and in the ms text (callouts)

SYNOPSIS IMAGE: missing, please provide.

SYNOPSIS TEXT: missing, please provide.

R&T TABLE: in but it is provided in the manuscript file, we need it removed and uploaded as a separate file (file format: Reagents table).

NOTES:

- Summary should be renamed to Abstract
- Structured Methods should be renamed to Methods

Partial reuse between Figure 8C and Table S1 Page 5. This should be mentioned and explained.

Figure EV4A - Top lane 12h. Appears blank currently - Please make sure to provide source data for this one.

P-values have to be provided in their exact values and not as " $p < 0.05$ ".

Comments from referees

Referee #1:

The paper is OK, but there are a few minor editorial issues that maybe addressed.

The referencing is funny, as in (Martin, Stanger-Hall et al., 2019) or (Ivanova, Nguyen et al., 2017, Wilts, Apeleo Zubiri et al.) - no publication year; but also (Blagodatski et al., 2015, Buscher, Kryuchkov et al., 2018).

Lepidoptera and Diptera (and Gyrinidae and Ascalaphidae) should not be in italics.

I think it should be: wild type *B. mandarina*

The Arrhenius equation is usually with a minus sign, but apparently the authors prefer a negative activation energy.

Referee #3:

This is a fine revision, but I am still not convinced that EMBO Reports is the right journal for this work. Scientifically, I don't see anymore shortcomings that need to be corrected, I just don't find this work interesting enough for a journal of this scope giving the little additional impact of this work.

Second round

Comments from editorial assistance team

- Conflict of Interest: included, but it needs to be renamed to Disclosure and Competing Interests Statement

- *Thank you for your comments. All changes from this round are highlighted in green. We have renamed this section to Disclosure and Competing Interests Statement.*

AC/CRedit: needs to be removed from the manuscript text and included only in our system.

- *We have removed the Authors Contributions section from the manuscript, as requested.*

REFERENCES: if a reference has more than 10 authors, et al should be used after the 10th name. Also, please see comments for Referee #1

- *We have corrected the formatting of all references accordingly.*

FUNDING INFO: OK, but the following grant should also be entered in the system in the grant field: E-COST-GRANT-CA21159-a405c60a.

- *We have updated the funding information and ensured that the grant is correctly entered in the system: European Cooperation in Science and Technology (COST), grant number CA21159 (p. 13).*

FIGURE CALLOUTS: missing callouts for - individual panel of Figure 1 A-D, Figure 4C, Appendix Table S5;

- *We have added the missing callouts for Figure 1A–D, Figure 4C, and Appendix Table S5 (pp. 3, 8, and 11).*

"supplementary" nomenclature should not be used. Please correct.

- *We have removed/renamed all uses of the “supplementary” nomenclature throughout the manuscript.*

APPENDIX FILE WITH Table of Content: in, but the Table of Content should list each item and its title page; the nomenclature is not OK, needs to be Appendix Table S1, etc, Appendix Figure S1, etc. throughout the Appendix file and in the ms text (callouts)

- *We have rebuilt the Appendix Table of Contents, listing each item individually with its corresponding title page, and updated all Appendix figure/table callouts accordingly.*

SYNOPSIS IMAGE: missing, please provide.

SYNOPSIS TEXT: missing, please provide.

- *We have provided both the synopsis image and the synopsis text.*

R&T TABLE: in but it is provided in the manuscript file, we need it removed and uploaded as a separate file (file format: Reagents table).

- *We have removed the R&T table from the manuscript and uploaded it as a separate file in the required format.*

NOTES:

- Summary should be renamed to Abstract
- Structured Methods should be renamed to Methods

- *We have renamed Summary to Abstract and Structured Methods to Methods.*

Partial reuse between Figure 8C and Table S1 Page 5. This should be mentioned and explained.

- *The partial reuse between Figure 8C and Appendix Table S1 Page 5 was unintentional. We have replaced the image in Table S1 while keeping the underlying information unchanged.*

Figure EV4A - Top lane 12h. Appears blank currently - Please make sure to provide source data for this one.

- *The previously missing source data for Figure EV4 A have now been uploaded.*

P-values have to be provided in their exact values and not as " $p < 0.05$ ".

- *All p-values have been updated to their exact numerical values.*

Referee #1:

The paper is OK, but there are a few minor editorial issues that maybe addressed.

The referencing is funny, as in (Martin, Stanger-Hall et al., 2019) or (Ivanova, Nguyen et al., 2017, Wilts, Apeleo Zubiri et al.) - no publication year; but also (Blagodatski et al., 2015, Buscher, Kryuchkov et al., 2018).

- *Thank you for these helpful observations. We have corrected the reference formatting throughout.*

Lepidoptera and Diptera (and Gyrinidae and Ascalaphidae) should not be in italics.

I think it should be: wild type *B. mandarina*

- *All Family and Order names have been changed to regular font, and B. mandarina has been corrected to “wild type Bombyx mandarina.”*

The Arrhenius equation is usually with a minus sign, but apparently the authors prefer a negative activation energy.

- *Thank you for pointing this out. For clarity of presentation, we have revised the Arrhenius equation and adjusted the activation energy accordingly.*

Mikhail Kryuchkov
University of Geneva
Department of Cell Physiology and Metabolism
Switzerland

Dear Dr. Kryuchkov,

I am very pleased to accept your manuscript for publication in the next available issue of EMBO reports. Thank you for your contribution to our journal.

You may qualify for financial assistance for your publication charges - either via a Springer Nature fully open access agreement or an EMBO initiative. Check your eligibility: <https://link.springer.com/journal/44319/how-to-publish-with-us>

Yours sincerely,

Yehu Moran
Academic Editor
EMBO Reports

>>> Please note that it is EMBO Reports policy for the transcript of the editorial process (containing referee reports and your response letter) to be published as an online supplement to each paper. If you do NOT want this, you will need to inform the Editorial Office via email immediately. More information is available here: <https://link.springer.com/partners/embo-press/editorial-policies#Peer%20review>